# GIGANTEA recruits the UBP12 and UBP13 deubiquitylases to regulate accumulation of the ZTL photoreceptor complex

Chin-Mei Lee [1], Man-Wah Li[1,2], Ann Feke[1,2], Wei Liu[1,2], Adam M. Saffer [1] & Joshua M. Gendron[1]

ZEITLUPE (ZTL), a photoreceptor with E3 ubiquitin ligase activity, communicates end-of-day light conditions to the plant circadian clock. It still remains unclear how ZTL protein accumulates in the light but does not destabilize target proteins before dusk. Two deubiquitylating enzymes, UBIQUITIN-SPECIFIC PROTEASE 12 and 13 (UBP12 and UBP13), which regulate clock period and protein ubiquitylation in a manner opposite to ZTL, associate with the ZTL protein complex. Here we demonstrate that the ZTL interacting partner, GIGANTEA (GI), recruits UBP12 and UBP13 to the ZTL photoreceptor complex. We show that loss of *UBP12* and *UBP13* reduces ZTL and GI protein levels through a post-transcriptional mechanism. Furthermore, a ZTL target protein is unable to accumulate to normal levels in *ubp* mutants. This demonstrates that the ZTL photoreceptor complex contains both ubiquitin-conjugating and -deconjugating enzymes, and that these two opposing enzyme types are necessary for circadian clock pacing. This shows that deubiquitylating enzymes are a core element of circadian clocks, conserved from plants to animals.

[1] Department of Molecular, Cellular, and Developmental Biology, Yale University, New Haven, CT 06511, USA. [2] These authors contributed equally: Man-Wah Li, Ann Feke, Wei Liu. Correspondence and requests for materials should be addressed to J.M.G. (email: joshua.gendron@yale.edu)

Circadian clocks in all organisms rely on photoreceptors to sense light and entrain the central oscillator. The exact timing of the light-to-dark transition (dusk) is especially important for plants, as this indicates the length of the day and provides seasonal timing information necessary for the adjustment of plant developmental processes[1–8]. One way that Arabidopsis senses the end of the day is by using a unique photoreceptor called ZEITLUPE (ZTL) to control the stability of circadian clock transcription factors differentially in the light and the dark[9]. ZTL contains an N-terminal light-oxygen-voltage sensing (LOV) domain, which senses blue light. Adjacent to the LOV domain are the F-box domain, which allows ZTL to function as an E3 ubiquitin ligase, and a Kelch-repeat domain. ZTL mediates degradation of transcription factors that are at the core of the plant circadian clock, including TIMING OF CAB2 EXPRESSION 1, PSEUDO-RESPONSE REGULATOR 5, and CCA1 HIKING EXPEDITION (TOC1, PRR5, and CHE)[10–15]. In the light, ZTL accumulates to high levels but is unable to mediate degradation of the clock transcription factors[16,17]. The accumulation of ZTL protein during the day is dependent on interaction with the co-chaperone protein GIGANTEA (GI)[18–20]. GI interacts with ZTL through the LOV domain in the light and dissociates from ZTL in the dark, allowing ZTL to mediate degradation of its target proteins and then be degraded by the ubiquitin proteasome system, likely through autocatalytic activity[10,11,16–18,21,22]. One of the roles of GI is to recruit HSP70/HSP90 for maturation of the ZTL protein in the light, but ZTL is unable to mediate ubiquitylation and degradation of target proteins until dark[10–12,19,23]. It was proposed that GI can promote maturation of ZTL and block or counteract ZTL activity; however, this second role for GI has not been investigated in depth[12,23].

We previously identified ZTL-interacting proteins using immunoprecipitation followed by mass spectrometry (IP-MS) with a "decoy" ZTL that lacks E3 ubiquitin ligase activity and stably binds interacting proteins[14]. We identified UBIQUITIN-SPECIFIC PROTEASE 12 and 13 (UBP12 and UBP13) as high-confidence ZTL-interacting proteins, which were shown previously to have an unspecified role in clock function[14,24]. UBP12 and UBP13 also interact with GI in IP-MS experiments[25], suggesting that either the UBPs interact with ZTL and GI independently or that ZTL, GI, and the UBPs exist together in a complex. UBP12 and UBP13 are closely related deubiquitylating enzymes that can cleave lysine 48-linked mono- or poly-ubiquitin from substrates[24,26], interestingly, a biochemical role opposite to that of ZTL. In addition to regulating the circadian clock, they are also involved in flowering time, pathogen defense, root differentiation, and hormone signaling[26–30].

Here, we show that UBP12 and UBP13 interact with the ZTL photoreceptor complex in a GI-dependent manner. Supporting this idea, genetic analyses show that UBP12 and UBP13 impact clock function through the same genetic pathway as ZTL and GI. Finally, we demonstrate that UBP12 and UBP13 are necessary for the proper daily accumulation of ZTL, GI, and TOC1 proteins. These results support the idea that in plants the communication of end-of-day light information relies on a photoreceptor complex that contains both ubiquitin conjugation activity and ubiquitin deconjugation activity.

## Results

**ZTL, GI, and UBP12/UBP13 form a trimeric complex.** Previously, it was shown that UBP12 and UBP13 associate with ZTL and GI in vivo[14,25]. To test whether UBP12 or UBP13 proteins interact with the members of the ZTL/GI protein complex we performed yeast two-hybrid assays. We found that UBP12 and

UBP13 interacted with GI but not with ZTL or the ZTL target proteins TOC1, PRR5, or CHE (Fig. 1a). We next tested the interaction between GI and UBP12 and UBP13 in planta via bimolecular fluorescence complementation (BiFC) in Arabidopsis protoplasts (Fig. 1b). GI, UBP12, and UBP13 are localized in the cytoplasm and nucleus[18,24], and our BiFC results show that UBP12 and UBP13 interact with GI in both compartments with strong signal in the nucleus and weaker but detectable signal in the cytoplasm. The interacting complexes of UBP12 and GI formed nuclear foci, similar to the localization of GI alone[31]. UBP12 and UBP13 contain a MATH-type (meprin and TRAF homology) protein interaction domain and a ubiquitin-specific protease (USP) domain (Supplementary Fig. 1). The MATH domains of UBP12 and UBP13 were necessary for interaction with GI while the protease domain and the C-terminal portions did not mediate GI-interaction (Fig. 1c). This suggests that the interaction between GI and UBP12 or UBP13 is not dependent on the UBP USP domains binding to poly-ubiquitylated GI protein.

We next determined whether GI was necessary to bridge the interaction between UBP12 or UBP13 and ZTL in vivo by performing IP-MS on wild-type (Col-0) and gi-2 mutant transgenic lines expressing the decoy ZTL protein (Supplementary Fig. 2). We collected samples at 9 h after dawn from plants grown in 12 h light/12 h dark cycles to capture the time when ZTL and GI are normally interacting. We found that UBP12 and UBP13 were enriched in the Col-0 samples (p-value = 3.58E-5 and 0.0113 for UBP12 and UBP13, respectively), but not in the gi-2 mutant (p-value = 1 for both) (Fig. 1d and Supplementary Data 1). These results indicate that GI is required for UBP12/UBP13 to form a complex with ZTL, substantiating our interaction studies in heterologous systems. Notably, LKP2, a known ZTL-interacting partner, associated with ZTL in the presence or absence of GI and suggests that the decoy ZTL is able to form biologically relevant protein complexes even in the gi-2 mutant[32]. Altogether these results suggest that the GI protein physically bridges the interaction between UBP12 or UBP13 and ZTL in vivo.

As a complementary approach to the IP-MS (Fig. 1d) we co-expressed FLAG-UBP12 or FLAG-UBP13 with HA-GI and Myc-ZTL in N. benthamiana leaves. We then performed immunoprecipitation with anti-FLAG antibody and detected the presence of FLAG-UBP12, FLAG-UBP13, HA-GI, and Myc-ZTL using western blotting (Fig. 1e). In the FLAG immunoprecipitation samples, HA-GI was always detected when co-expressed with FLAG-UBP12 or FLAG-UBP13, showing that UBP12 and UBP13 interact with GI independently of the presence of Myc-ZTL. Furthermore, Myc-ZTL was undetectable in the FLAG immunoprecipitation samples unless co-expressed with HA-GI showing that the interaction between UBP12 or UBP13 and ZTL is dependent on GI. These assays support our previous results (Fig. 1a–d) and show that a trimeric complex between full-length ZTL, GI, and UBP12 or UBP13 can form in vivo (Fig. 1f).

**UBP12/UBP13 are in the same genetic pathway as ZTL and GI.** Our physical interaction model (Fig. 1f) led us to hypothesize that UBP12 and UBP13 regulate the circadian clock through the same genetic pathway as ZTL and GI. We tested this via epistasis analyses with loss-of-function mutants in ZTL, GI, UBP12, and UBP13. Previously, it was shown that knockdown of UBP12 and UBP13 results in shortened clock periods[24]. We first determined the period of a series of mutant alleles in UBP12 and UBP13 by crossing them to the pCCA1::LUC clock reporter transgenic line and measuring luciferase activity (Fig. 2a–d). We found that single mutations in either UBP12 or UBP13 shortened the clock

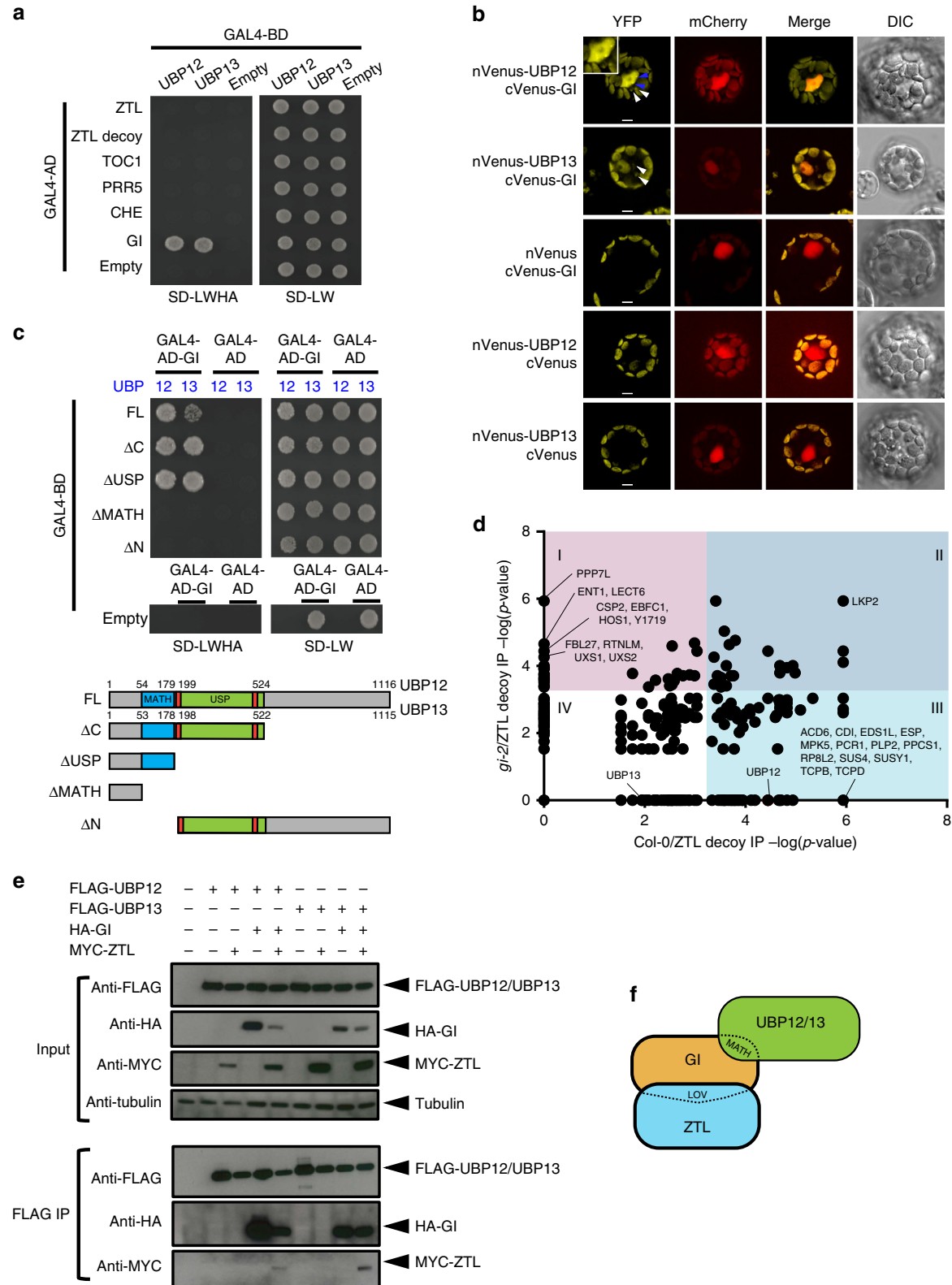

period with period lengths that varied from 0.4 to 1 h shorter than wild type. We next generated *ubp12-1/gi-2* and *ubp13-1/gi-2* double mutants and measured the expression of the core clock gene *CCA1* during a 2-day circadian time course in constant light using quantitative reverse-transcription PCR (qRT-PCR) (Fig. 2e, f and Supplementary Table 1). LS Periodogram analysis using the Biodare2 platform [biodare2.ed.ac.uk[33]] showed that the *ubp12-1/*

*gi-2* double mutant had a similar phase and amplitude of *CCA1* expression to the *gi-2* mutant alone and a period more similar to *ubp12-1* (Supplementary Table 2). These results show a non-additive interaction and suggest they function in the same circadian genetic pathway. The *ubp13-1/gi-2* double mutant had a similar amplitude to the *gi-2* mutant but had a more similar phase and period to the *ubp13-1* mutant (Supplementary Table 2). This

**Fig. 1** GI bridges the interactions between ZTL and UBP12 or UBP13. **a** Yeast two-hybrid showing interaction between GI and UBP12 or UBP13. The GAL4 DNA-binding domain (GAL4-BD) fused to UBP12 or UBP13 and either ZTL variants (ZTL and ZTL decoy), ZTL targets (TOC1, PRR5, and CHE) or GI fused to GAL4 activation domain (GAL4-AD) were grown on SD-LW medium for autotrophic selection and on SD-LWHA medium to test for interactions. **b** Bimolecular fluorescence complementation (BiFC) assays to examine the interactions of UBP12 or UBP13 and GI fused to the N- or C-terminus of Venus (YFP) were performed in *Arabidopsis* protoplasts. The blue arrows indicate the interacting complex forming nuclear foci. The white arrows show fluorescence signal in the cytoplasm. mCherry-VirD2NLS was co-expressed as a nuclear marker, and the scale bar indicates 10 μm. **c** The protein domains of UBP12 and UBP13 required to interact with GI were tested using yeast two-hybrid assays. The full-length (FL) or truncated UBP12 or UBP13 fragments as diagramed in the lower portion of the panel were fused to GAL4-BD to test for interaction with GAL4-AD-GI. **d** Scatter plot of proteins identified by IP-MS of ZTL decoys in the Col-0 and *gi-2* genotypes. The significance of the interactions were evaluated by SAINTexpress (see Methods and Supplementary Data 1 for complete information) with a false discovery rate (FDR) cutoff < 0.01 and *p*-value ≤ 5.37E-4 to separate interacting proteins into four groups. Group I: significant interactions with ZTL decoy in the *gi-2* but not Col-0. Group II: significant interactions with ZTL decoy in both Col-0 and *gi-2*. Group III: significant interactions with ZTL decoy in the Col-0 but not *gi-2*. Group IV: Non-significant interactions with ZTL decoy in both Col-0 and *gi-2*. The interacting proteins significantly enriched in the *gi-2* mutant over Col-0 were labeled along the *y*-axis, and the proteins enriched in the Col-0 over the *gi-2* mutant were labeled along the *x*-axis. **e** Co-immunoprecipitation assays showing that UBP12 or UBP13 interact with ZTL in a GI-dependent manner. FLAG-UBP12 or FLAG-UBP13 were co-infiltrated with HA-GI and Myc-ZTL in *Nicotiana benthamiana* leaves. Anti-FLAG antibody was used to immunoprecipitate FLAG-UBP12 or FLAG-UBP13. Western blotting with anti-FLAG, anti-HA, or anti-Myc was used to detect the presence of FLAG-UBP12, FLAG-UBP13, HA-GI, or Myc-ZTL in the immunoprecipitated samples and inputs. **f** The diagram depicts the interaction between GI and the MATH domain of UBP12 or UBP13, and between GI and the LOV domain of ZTL. The source data are provided as a Source Data file. Blot images were cropped from their original size, which can be found in Source Data file

---

again shows a non-additive genetic interaction but also suggests that the roles of *UBP12* and *UBP13* have slightly diverged with respect to clock function. We also crossed the *gi-2* mutant with the *ubp12-2w* mutant, which had reduced expression of both *UBP12* and *UBP13* and the shortest clock period of the tested *ubp* mutant alleles (Supplementary Fig. 3 and Fig. 2a–d). The pattern of *CCA1* expression in the *ubp12-2w/gi-2* double mutant was nearly identical to the *gi-2* mutant, further confirming that the effects of the *UBP*s and *GI* are not additive (Supplementary Table 2). These results indicate that *UBP12* and *UBP13* work in the same pathway as *GI* to control clock function.

*ZTL* functions downstream of *GI* to regulate the circadian clock[17]. Thus, we hypothesized that *ZTL* would function downstream of *UBP12* and *UBP13* as well. To test the genetic interaction between *UBP12* or *UBP13* and *ZTL*, we crossed *ubp12-1* and *ubp13-1* to the *ztl-4* null mutant (Fig. 2g, h). The daily expression patterns of *CCA1* in the *ubp12-1/ztl-4* and *ubp13-1/ztl-4* double mutants were nearly identical to the *ztl-4* mutant alone in phase and amplitude (Supplementary Table 2). Interestingly, the period data showed that the *ubp12-1/ztl-4* was more similar to *ztl-4* than *ubp12-1*, but the *ubp13-1/ztl-4* is more similar to *ubp13-1*. These data suggest that *ZTL* is epistatic to *UBP12* and *UBP13* but that *UBP13* has diverged in function from *UBP12*. It is important to note that the qRT-PCR data are below the suggested resolution for Biodare2 analysis, which can result in inaccurate period calls (i.e., *ubp13-1* period is estimated by Biodare2 as the same period as wild type in this experiment). These results corroborate our physical interaction studies and suggest that *UBP12* and *UBP13* regulate the circadian clock upstream of *ZTL*.

**UBP12 deubiquitylase activity is required for clock function.** UBP12 and UBP13 are functional deubiquitylases that can cleave poly-ubiquitin from generic substrates[24,26]. We tested whether this deubiquitylation activity is necessary for their role in circadian clock function. To do this, we performed complementation studies with wild-type *UBP12* and mutant *UBP12^C208S*. UBP12^C208S has a mutation in the cysteine-box of the USP enzymatic core (Supplementary Fig. 1) that renders it non-functional as a deubiquitylase[24,27,28]. We transformed *UBP12-YFP* or *UBP12^C208S-YFP* driven by the *UBP12* native promoter into the *ubp12-1* mutant and analyzed a population of T1 transgenic lines. In this experiment we consider a line to have rescued the *ubp12-1* mutant clock phenotype if it has a period length longer than the

average period length of the *ubp12-1* plus one standard deviation. Using this criteria, 10 of 32 transgenic lines (31%) transformed with catalytically active UBP12 rescued the short period defect of the *ubp12-1* mutant. Strikingly, only one transgenic line transformed with the inactive UBP12^C208S was able to rescue the short period phenotype of *ubp12-1* (Fig. 2i, j). As reference, ~13% of the *ubp12-1* plants themselves and 62% of the wild-type plants fell into the rescue category. This is likely due to normal variations in population level data of this type. We further confirmed that UBP12-YFP and UBP12^C208S-YFP were both localized to the cytoplasm and nucleus (Supplementary Fig. 4a), and that there is no observable effect of the C208S mutation on UBP12 stability (Supplementary Fig. 4b, c) when these proteins are transiently expressed. This suggests that differences between the wild type and C208S variants of UBP12 are not due to mislocalization or altered accumulation of the UBP12^C208S protein. Altogether, these results indicate that the deubiquitylating functions of UBP12 are necessary for its role in regulating the circadian clock.

**UBP12/UP13 stabilize GI, ZTL, and TOC1.** By cleaving poly-ubiquitin from proteins, deubiquitylase enzymes can regulate protein stability and accumulation[28,30,34,35]. The physical and genetic interactions shown for UBP12, UBP13, GI and ZTL prompted us to hypothesize that the UBP12 and UBP13 regulate GI or ZTL protein levels, allowing for accumulation of the proteins in the end of the day. We measured the level of HA-tagged GI under the control of the GI native promoter (*pGI::GI-HA*) in the *ubp12-1* and *ubp13-1* mutants during a 12 h light/12 h dark time course (Fig. 3a). GI protein levels were ~50% lower in the *ubp12-1* and *ubp13-1*. Messenger RNA (mRNA) expression of *GI-HA* was also ~25% lower than wild type at the peak of *GI* mRNA expression, ZT8 (Fig. 3b). This suggests that GI protein accumulation is partially dependent on UBP12 and UBP13, but that altered transcription of *GI* could also have an effect on GI protein.

Next, we measured ZTL protein levels in the *ubp12-1* and *ubp13-1* mutants (Fig. 3c). ZTL protein levels were substantially decreased in the *ubp12-1* and *ubp13-1* mutants throughout the entire day/night cycle. Overexposure of the immunoblot showed that a small amount of ZTL protein can still accumulate in the *ubp* mutants (Fig. 3c). The expression of *ZTL* mRNA was largely unaffected in these lines (Fig. 3d), suggesting that the decrease in ZTL protein levels was caused by a post transcriptional mechanism. This is similar to the post transcriptional control of ZTL reported in *gi* loss-of-function mutants[17], and indicates

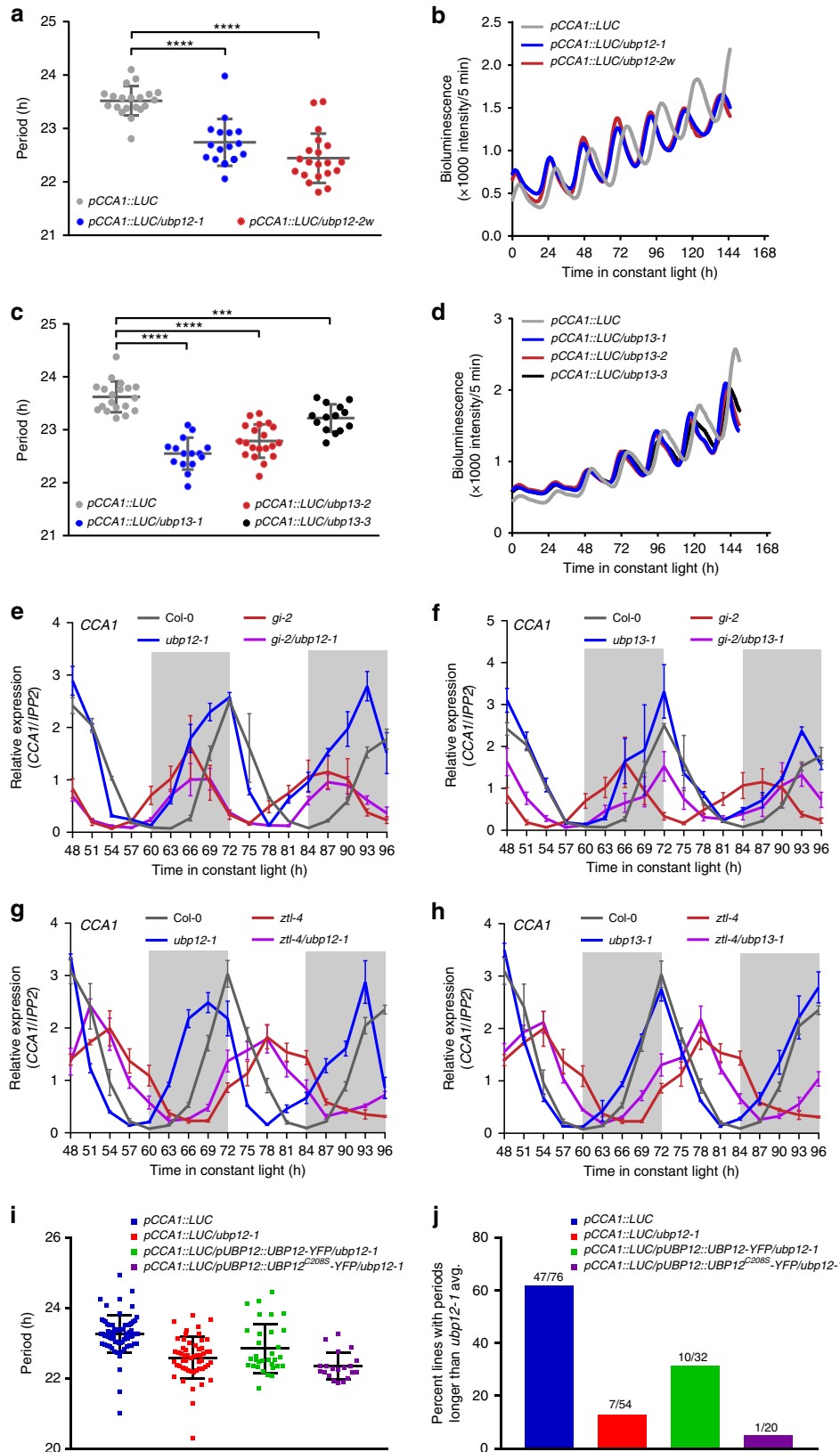

that UBP12 and UBP13 are necessary for robust accumulation of the ZTL protein.

Interestingly, the *ubp12-1* and *ubp13-1* mutants caused severe reduction in the levels of the ZTL protein but had a short period phenotype, opposite to the long period phenotype of *ztl* loss-of-function mutants. Normally, loss of ZTL causes aberrantly high

levels of TOC1 protein while overexpression of ZTL causes low levels of TOC1 protein[10,11,23,36]. To determine if UBP12 and UBP13 affect TOC1 protein levels, we crossed a transgenic line expressing TOC1 fused to YFP under the *TOC1* promoter (*TMG*) to the *ubp12-1* and *ubp13-1* mutants and measured TOC1 protein levels (Fig. 3e). TOC1 protein levels were severely reduced in the

**Fig. 2** *UBP12* and *UBP13* regulate the circadian clock through the same pathway as *GI* and *ZTL*. **a-d** The *ubp12* and *ubp13* mutants have short period phenotypes. **a, c** The periods of circadian marker *pCCA1:Luciferase* (*pCCA1::LUC*) in the wild type (Col-0) (*n* = 20 for **a** and *n* = 19 for **c**), *ubp12-1* (*n* = 16), *ubp12-2w* (*n* = 20), *ubp13-1* (*n* = 15), *ubp13-2* (*n* = 20), and *ubp13-3* (*n* = 14) were measured with bioluminescent assays. Each symbol represents the period from one seedling, and the average period and standard deviation are labeled with gray bars. The significance of period changes between wild type and mutants were analyzed with a two-tailed Welch's *t*-test (*** for *p*-value < 0.001; **** for *p*-value < 0.0001). Three biological replicates were performed with similar results, and one dataset is presented. **b, d** The average bioluminescence of the lines displayed in **a** and **c** were plotted against time after transfer from 12 h light/12 h dark entrainment conditions to constant light. **e, f** Circadian expression of *CCA1* in Col-0, *ubp12-1*, *ubp13-1*, *gi-2*, *gi-2/ubp12-1*, and *gi-2/ubp13-1* after transferring to constant light for 48 h from the entrainment conditions was measured using qRT-PCR. Subjective dark is colored with light gray. The data represent the average relative expression of *CCA1* normalized to *IPP2* from three biological replicates, and the error bars are the standard deviation. The same Col-0 and *gi-2* data were plotted twice (in **e** and **f**) for clarity in the data presentation and for comparison with the other mutant lines. **g, h** The circadian expression of *CCA1* in Col-0, *ubp12-1*, *ubp13-1*, *ztl-4*, *ztl-4/ubp12-1*, and *ztl-4/ubp13-1* after transferring to constant light for 48 h from the entrainment conditions was measured using qRT-PCR. The data analyses and presentation are the same as **e-f**. The same Col-0 and *ztl-4* data were plotted twice (in **g** and **h**). **i** The circadian period of *pCCA1::LUC* in the wild type (*n* = 76), *ubp12-1* (*n* = 54), *ubp12-1* mutant complemented with *pUBP12::UBP12-YFP* (*n* = 32) or deubiquitylating activity-dead *pUBP12::UBP12CS-YFP* (*n* = 20). Each symbol represents the period from one seedling, and the black bars indicate the average period and standard deviation. The wild type and *ubp12-1* mutants are homogenous populations, and the complementation lines are individual T1 transgenic lines. The presented data are from three independent biological replicates. **j** Quantitation of the number of lines, from panel **i**, with periods greater than the average of the *ubp12-1* mutant plus one standard deviation. The source data are provided as a Source Data file

*ubp12-1* and *ubp13-1* mutants while mRNA expression of the *TOC1-YFP* transgene was similar in the wild type and mutant backgrounds, suggesting that the decrease in TOC1 protein levels was caused by a post-transcriptional mechanism (Fig. 3f). Notably, TOC1 protein was unable to accumulate to high levels in the light in the *ubp* mutants (Fig. 3e at 12 h after dawn). This is similar to the effects of the *gi-2* mutant, where TOC1 protein levels never accumulate to full wild-type levels[17]. This suggests that the period effects of the *ubp12* and *ubp13* mutants may be caused by the same mechanism as the short period of the *gi-2* mutant.

## Discussion

We have shown that UBP12 and UBP13 are components of the ZTL-GI photoreceptor complex that are necessary for accumulation of the proteins in the end of the day. UBP12 and UBP13 can remove poly-ubiquitin from targets non-specifically[24,26]. Thus, we hypothesize that UBP enzymes are recruited by GI to the ZTL photoreceptor complex to prevent formation of poly-ubiquitin chains, resulting in increased stability of the protein complex (Fig. 4). Interestingly, ZTL protein levels were severely damped in the *ubp12* and *ubp13* mutants, but counterintuitively the ZTL target, TOC1, also had reduced levels (Fig. 3c–f). This effect is similar to what was observed in a *gi* loss-of-function mutant, and suggests that GI and UBP12 and UBP13 can counterbalance the activity of ZTL during the day, allowing TOC1 to accumulate to high levels before being degraded[17]. Although ZTL levels were decreased in the *ubp* mutants, there was still a small amount that could potentially decrease TOC1 levels in the light (Fig. 3c long exposure). This is different than what was seen when HSP90 activity was inhibited, resulting in lower ZTL levels but higher TOC1 levels. This suggests that HSP90 is necessary for ZTL protein maturation and to promote its activity[22]. These data in combination with our results suggest that GI performs two roles in the ZTL photoreceptor complex: (1) acting as a co-chaperone that recruits HSP proteins to facilitate ZTL maturation[19,20], and (2) counterbalancing the role of ZTL in ubiquitin conjugation with UBP12 and UBP13 present to deconjugate ubiquitin. The light-regulated nature of the ZTL–GI interaction also indicates that light is controlling the balance of ubiquitin conjugation and deconjugation that allows the ZTL photoreceptor complex to accurately degrade proteins at the correct time of day. It was previously shown that mammalian and insect circadian clocks utilize deubiquitylation to regulate stability and subcellular localization of clock proteins[37–39]. In light of this, our results further demonstrate that deubiquitylation activity is an evolutionarily conserved feature of the clocks of higher eukaryotes Furthermore, the mammalian ortholog of UBP12 and UBP13, USP7, impacts clock function in response to environmental stress[40,41], suggesting that these deubiquitylases are conserved clock regulators across evolution.

## Methods

**Plant materials and growth conditions.** The *Arabidopsis* seeds of Col-0, *ubp12-1* (CS423387), *ubp12-2w* (CS2103163), *ubp13-1* (SALK_128312), *ubp13-2* (SALK_024054), *ubp13-3* (SALK_132368)[24], *gi-2* (cs3370)[42,43], *ztl-4* (SALK_012440)[44], *pGI::GI-HA* (CS66130)[45], and TMG (CS31390)[10] were described previously and obtained from ABRC. The *ubp12-1/gi-2*, *ubp12-2w/gi-2*, *ubp13-1/gi-2*, *ubp12-1/ztl-4*, and *ubp13-1/ztl-4* double mutants were generated by crossing and genotyped by PCR. The *pGI::GI-HA* and TMG lines were crossed to *ubp12-1* and *ubp13-1*, and the homozygous lines were selected by genotyping and gentamycin resistance.

For IP-MS, the *35S::FLAG-His-ZTL-decoy* transgenic lines and *35S::FLAG-His-GFP* control were described previously[46], and the same constructs were transformed into the *gi-2* background by floral-dip method[47].

For the bioluminescent assays, the circadian reporter line *pCCA1::Luciferase* (*pCCA1::LUC*)[48] was crossed to the *ubp12* and *ubp13* mutants. The *pUBP12::UBP12-YFP* variants (see Cloning section) were transformed into *pCCA1::LUC/ubp12-1* by floral-dip[47] for complementation experiments.

For growth conditions of *Arabidopsis* seedlings, the seeds were surface sterilized with ethanol, cold stratified, plated on ½ strength MS (Murashige and Skoog medium, Caisson Laboratories, MSP01) medium with 0.8% Agar (AmericanBio, AB01185), and grown at 22 °C under 12 h light/12 h dark as described previously[46] unless specified otherwise. For soil-grown conditions, plants were grown in Fafard-2 mix under 16 h light/8 h dark at 22 °C.

For circadian experiments, seedlings were grown on ½ strength MS medium under 12 h light/12 h dark at 22 °C for 10 days, transferred to continuous light (LL) at 22 °C for 48 h before starting harvest. For the 12 h light/12 dark (LD) experiments, 12-day-old seedlings grown on ½ strength MS medium were used.

**Cloning.** The GATEWAY pENTR™/D-TOPO entry vectors (Thermo Fisher Scientific, K240020) of ZTL full-length, ZTL decoy, CHE, TOC1, and PRR5 were obtained from previous reports[46,48,49]. For GI, UBP12 and UBP13, the full-length coding regions were amplified from complementary DNA by PCR and cloned into pENTR™/D-TOPO vectors. These entry clones were then sub-cloned into GATEWAY compatible yeast two-hybrid vectors (pGADT7-GW and pGBKT7-GW)[50] or BiFC vectors (pUC-DEST-VYCE®GW and pUC-DEST-VYNE®GW)[51] with GATEWAY recombination cloning (Thermo Fisher Scientific).

To construct the fragments of UBP12 and UBP13 into yeast two-hybrid pGADT7-GW vectors, the desired fragments were first amplified from the full-length UBP12 or UBP13 entry vectors by PCR and cloned into pENTR™/D-TOPO vectors before being sub-cloned into pGADT7-GW with GATEWAY cloning.

For the UBP12 complementation plasmids, the pENTR™/D-TOPO-UBP12-NS vector served as template for site-directed mutagenesis to introduce a Cys to Ser mutation at a.a. 208 position using Q5® Site-Directed Mutagenesis Kit (NEB, E0554). Subsequently, UBP12-NS and UBP12C208S-NS in the pENTR™/D-TOPO entry vectors were sub-cloned into a modified GATEWAY compatible pGreenBarT vector[51] with 1.7 k bp upstream of ATG of UBP12 promoter region in the KpnI/XhoI sites. The primers used for cloning were listed in Supplementary Table 1.

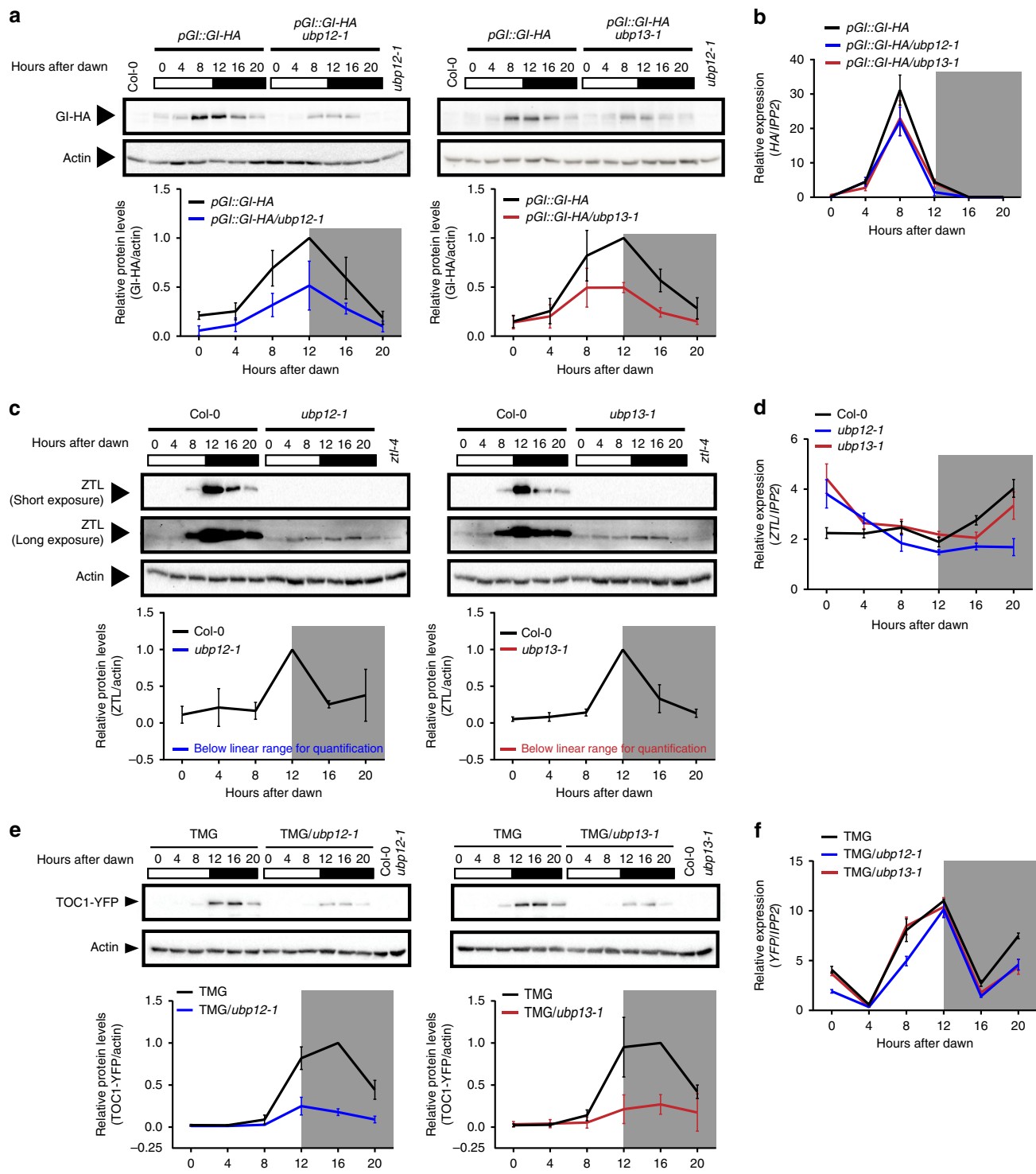

**Fig. 3** ZTL, GI, and TOC1 protein levels are regulated by UBP12 and UBP13. **a**, **c**, **e** The protein levels of HA-tagged GI driven by native promoter (*pGI::GI-HA*), ZTL and YFP-tagged TOC1 driven by the TOC1 promoter (TOC1 minigene or TMG) in the wild type (Col-0), *ubp12-1*, or *ubp13-1* mutants under diurnal conditions (12 h light/12 h dark) were detected by immunoblotting. The samples from 0 h to 12 h after dawn were harvested in light, and the samples from 16 h and 20 h after dawn were harvested in the dark (indicated by gray shading). The relative protein levels were quantified by normalization to actin. The Col-0 or *ztl-4* samples were used as negative controls for the antibodies. Plots represent the average protein levels from three biological replicates, and the error bars represent standard deviation. Compared to wild type, the levels of ZTL proteins in the *ubp12-1* and *ubp13-1* were below the linear range for quantification. In the *ztl-4* sample, the anti-ZTL antibody recognizes a non-specific band close to the size of endogenous ZTL in the long-exposure blots. **b**, **d**, **f** The relative mRNA levels of *GI-HA*, *ZTL*, or *TOC1-YFP* from the same time course samples were measured by qRT-PCR. The source data are provided as a Source Data file. Blot images were cropped from their original size, which can be found in Source Data file

 

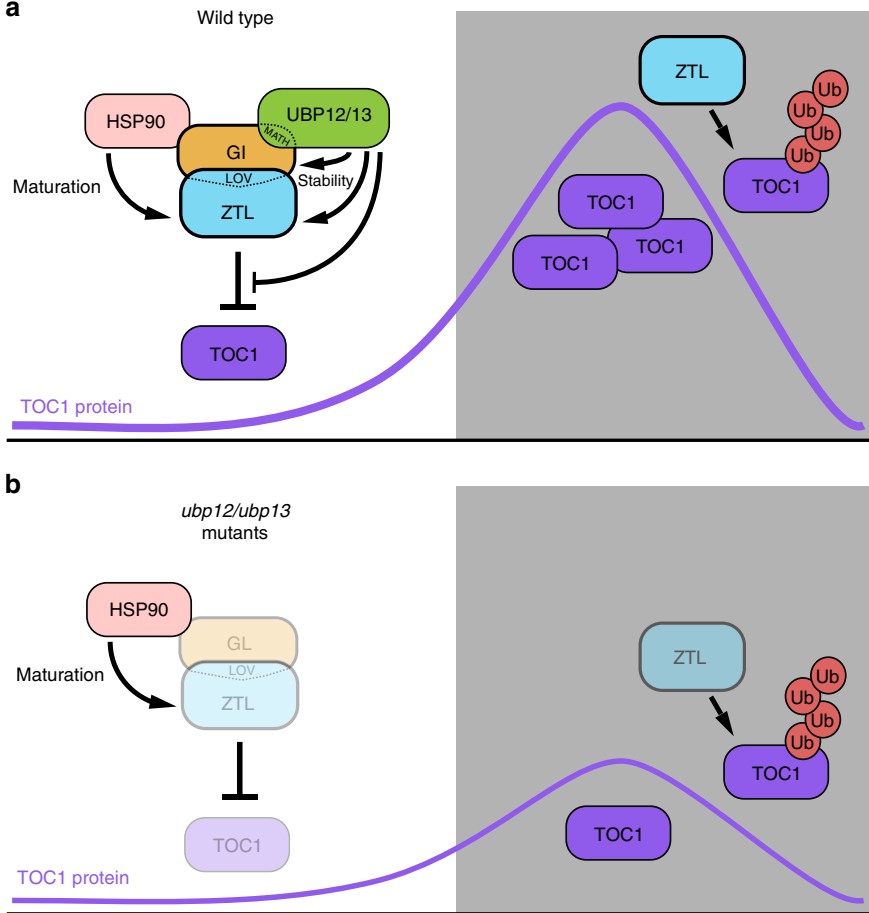

**Fig. 4** The proposed model for UBP12/UBP13 regulation of ZTL. **a** In the light, GI interacts with ZTL and acts as a co-chaperone, recruiting HSP90 to facilitate folding and maturation of the ZTL protein. Additionally, GI physically bridges an interaction between ZTL and UBP12 or UBP13. UBP12 or UBP13 stabilize the GI-ZTL protein complex before dusk. In the dark, GI dissociates from ZTL, and ZTL mediates ubiquitylation and degradation of the TOC1 protein. **b** Loss of *UBP12* or *UBP13* causes instability of ZTL and GI. Interestingly, the TOC1 protein levels are also reduced by loss of *UBP12* or *UBP13*, mimicking the *gi* loss-of-function mutant

**Yeast two-hybrid**. ZTL, ZTL decoy, GI, TOC1, PRR5 and CHE were fused to the GAL4-BD in pGBKT7-GW vectors, and the full-length or fragments of UBP12 and UBP13 were fused to the GAL4-AD in pGADT7-GW vectors by GATEWAY cloning. The interactions were tested on synthetic dropout medium as described previously[46].

**BiFC and confocal microscopy**. The coding region of GI, UBP12, or UBP13 in the GATEWAY entry vectors were cloned into protoplast GATEWAY destination vectors pUC-DEST-VYCE®GW and pUC-DEST-VYNE(R)GW[51], respectively, for transient transfections into protoplasts. pSAT6-mCherry-VirD2NLS was used as a nuclear marker. The protoplasts were isolated from 3- to 4-week-old *Arabidopsis* (Col-0) grown at 22 °C and transfected following the protocol of tape-Arabidopsis sandwich method[52]. After 14–18 h incubation in low-light conditions, protoplasts were imaged on a Nikon Ti microscope with using a $60 \times 1.4$ NA plan Apo objective lens as described previously[53]. The images were analyzed with FIJI[54].

**Immunoprecipitation and mass spectrometry (IP-MS)**. For the ZTL decoys in Col-0 background, homozygous *35S::FLAG-His-ZTL-decoy* transgenic lines along with Col-0 and *35S::FLAG-His-GFP* controls were used. For the ZTL decoys in the *gi-2* background, three independent T2 transgenic lines of *35S::FLAG-His-ZTL-decoy/gi-2* and *35S::FLAG-His-GFP/gi-2* were selected on ½ strength MS plates with 15 μg/ml ammonium glufosinate before being transferred to soil. Twenty-one-day-old soil-grown plants were entrained in 12 h light/12 h dark at 22 °C for 7 days prior to harvest. Leaf tissues were collected at 9 h after dawn for subsequent IP-MS. One-step IP-MS and MS spectral analyses were carried out as documented[46] with minor changes. The MS/MS spectral were searched against the SwissProt_2017 tax: Arabidopsis thaliana (thale cress) database (February 2017) using MASCOT MS/MS ion search engine version 2.6.0[55] with the following parameters: up to two missed cleavages; variable modifications included Acetyl (K), GlyGly(K), Oxidation (M), Phospho (ST), Phospho (Y); peptide tolerance ± 10 ppm; MS/MS tolerance ±

5 Da; peptide charge2 + and 3+. The protein lists identified by MASCOT were first filtered out non-specific interactions by removing proteins only present in the controls (Col-0, gi-2, 35S::FLAG-His-GFP/Col-0 and 35S::FLAG-His-GFP/gi-2). The SAINTexpress algorithm[56,57] were further performed to determine the significance of protein–protein interactions.

**Bioluminescent assays**. The *Arabidopsis* seedlings bearing *pCCA1::LUC* in the wild type (Col-0), *ubp12*, or *ubp13* mutants were grown in ½ strength MS medium and entrained in 12 h light/12 h dark for 7 days prior to being transferred to new ½ strength MS plates and constant light (LL) for circadian free-run experiments. For the various *pUBP12::UBP12-YFP* complementation T1 lines in the *pCCA1::LUC/ubp12-1* background, seedlings were first screened and entrained on the ½ strength MS plates containing 7.5 mg/ml ammonium glufosinate prior to being transferred to ½ strength MS medium and LL. The measurement of luciferase activities and analyses were described as previously[46].

**Real-time quantitative reverse-transcription PCR (qRT-PCR)**. RNA extraction, reverse-transcription, and constitution of qPCR reactions were followed as described previously[46], except for minor modifications. Four-hundred nanograms of total RNA were used for reverse-transcription reactions. For semi-quantification of gene expression, *IPP2* (AT3G02780) was used as an internal control. The relative expression represents means of $2^{(-\Delta CT)}$ from three biological replicates, in which $\Delta CT = (CT$ of Gene of Interest $- CT$ of $IPP2)$. The primers were listed in Supplementary Table 1.

**Immunoblotting**. The procedure of protein extraction from *Arabidopsis* seedlings, separation, detection with antibodies, and quantification are described as previously[46], except 60 μg total protein were used for immunoblotting. The primary antibodies used for detection are: for GI-HA, anti-HA-Biotin antibody (1:1000, 12158167001, Millipore-Sigma); for ZTL, anti-ZTL antibody[16] (1:200); for TMG,

anti-GFP (1:10000, ab-290, Abcam); for FLAG-ZTL decoy, anti-FLAG antibody (1:1000, F7425, Millipore-Sigma). To quantify expression levels, the levels of target proteins were normalized to actin (anti-Actin antibody, 1:2000, SAB4301137, Millipore-Sigma).

**Transient expression and confocal microscopy**. UBP12-NS and UBP12C208S-NS in the pENTR™/D-TOPO vectors were sub-cloned into inducible GATEWAY destination pABindGFP vectors[58] and transformed into the *Agrobacterium tumefaciens* strain GV3101 for transient expression in *Nicotiana benthamiana*. The *Agrobacterium* culture of pABindGFP-UBP12 or pABindGFP-UBP12C208S and the nuclear marker pABindcherry-AS2[59] were pelleted and resuspended in the infiltration solution (5% (w/v) Sucrose, 450 μM acetosyringone and 0.01% (v/v) Silwet). The bacterial infiltration solution was incubated at 4 °C for 2 h before infiltrated into 5-week-old *Nicotiana benthamiana* leaves. After 20 h of infiltration, the protein expression was induced by spraying leaves with 20 μM β-estradiol in 0.1% Tween 20. The leaves were imaged after 18 h of induction.

The leaf samples were imaged on a Zeiss LSM510 confocal microscope with a Plan-Apochromat 40 × /1.3 Oil objective. GFP was excited using 488 nm Argon laser and observed through a 505/530 nm bandpass filter. mCherry was excited using 543 nm HeNe laser and observed through a 585/615 nm bandpass filter. The images were processed with FIJI[54].

**co-IP in *Nicotiana benthamiana***. The full-length coding sequences of ZTL, GI, UBP12, and UBP13 in the pENTR™/D-TOPO vectors were sub-cloned into pEarlygate203, pEarlygate201 and pEarlygate202 plant binary vectors[60], respectively, and transformed into *Agrobacterium tumefaciens* strain GV3101. Agro-infiltration into *Nicotiana benthamiana* leaves was described in the previous section. In this co-immunoprecipitation experiment co-infiltration with P19 in the EHA105 *Agrobacterium* strain was used to increase expression of the transgenes. The leaf samples were harvested after 48 h of infiltration and snap frozen with liquid nitrogen. Protein extraction and co-immunoprecipitation with Anti-FLAG® M2 Magnetic Beads (M8823, Millipore-Sigma), and a one-step IP protocol was used as described previously[46,61]. The inputs and IP samples were resolved on NuPAGE 4–12% Bis-Tris Protein Gels (NP0321, Thermo Fisher Scientific) for immunoblotting. The primary antibodies used for detection are: for MYC-ZTL, anti-MYC antibody (1:10000, C3956, Millipore-Sigma); for HA-GI, anti-HA antibody (1:5000, H3663, Millipore-Sigma); for FLAG-UBP12 and FLAG-UBP13, anti-FLAG antibody (1:5000, F1804, Millipore-Sigma); for loading control, anti-tubulin antibody (1:5000, T5168, Millipore-Sigma).

**Reporting summary**. Further information on research design is available in the Nature Research Reporting Summary linked to this article.

## Data availability
The mass spectrometry proteomics data was deposited to the ProteomeXchange Consortium via the PRIDE partner repository[62] (https://www.ebi.ac.uk/pride/archive/). It is accessible via identifier PXD014636. The source data for Figs. 1, 2, and 3 and Supplementary Figs. 3 and 4 are in the Source Data file. Additional data and materials reported in this study are available from the corresponding author upon request.

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

## Acknowledgements

We thank Dr. David E. Somers for providing the anti-ZTL antibody. We would like to thank Dr. Nicole Clay and Dr. Jimi Miller for kindly sharing pABind vectors and the assistance with transient expression in the *Nicotiana benthamiana* experiments, the Keck Proteomics Facility at Yale for processing samples and analyzing proteomic data, Dr. Shirin Bahmanyar, Dr. Marshall Delise, and Dr. Joseph Wolenski for assistance with confocal microscopy experiments. We would also like to thank Suyuna Eng Ren, Chris Adamchek, Catherine Chamberlin, Chris Bolick, Christine Ventura, Denise George, and Sandra Pariseau for technical and administrative support. We would like to thank Dr. Vivian Irish, Dr. Mark Hochstrasser, and Dr. Eric Bennet for their helpful comments and insight. This work was supported by NSF EAGER grant 1548538 (J.M.G.), NIH R35GM128670 (J.M.G.), Rudolph J. Anderson Fund Fellowship (C.M.L.), Forest B.H. and Elizabeth D.W. Brown Fund Fellowship (C.M.L. and W.L.), NIH GM007499 (A.F.), The Gruber Foundation (A.F.), and NSF GRFP DGE-1122492 (A.F.).

## Author contributions

J.M.G. and C.M.L. conceived of the project. C.M.L., M.W.L., A.F., W.L. and A.M.S. conducted the experiments and analyzed the data. J.M.G., A.F. and C.M.L. wrote the manuscript.

## Additional information

**Competing interests:** The authors declare no competing interests.

