## [Peer Review File · Nature Communications]

Reviewers' comments:

Reviewer #1 (Remarks to the Author):

Lee et al. provide a detailed and compelling study regarding the role of UBP12/13 in regulating the stability and function of ZTL in the circadian clock. The data presented address a major question in the field; namely, how does ZTL accumulate to high levels in the light without destabilizing and targeting its substrates for degradation? To address this question, they show that UBP12 and 13 are part of the ZTL-GI complex. More specifically, GI is required to bring UBP12/13 to the ZTL-GI complex where it can function as a deubiquitylase to stabilize both ZTL and its degradation targets. They demonstrate that the interaction is specifically through GI in both yeast-2-hybrid assays, with confirmation *in vivo* using IP-MS. They are able to further validate that UBP12/13 function in the same pathway as GI and that ZTL is epistatic to UBP12/13. The combined results provide a complete biochemical and genetic model of how ZTL targets proteins for degradation in a time of day specific manner, whereby ZTL activity is modulated by a ZTL-GI-UBP complex where UBP12/13 functions antagonistically to ZTL to create a balance between ubiquitylation and de-ubiquitylation of ZTL and its targets. In so doing they also resolve an oddity in the circadian clock community regarding how TOC1 levels are low when ZTL is destabilized in either GI or UBP12/13 mutants.

Although I cannot address technical aspects regarding the genetic approaches used in this manuscript (and leave that for other reviewers), the biochemical and molecular approaches outlined in this manuscript are well conducted and compelling. Overall the manuscript is well written and is of high-significance to the field. For these reasons I support publication of the manuscript in *Nature Communications* as is.

If forced to make some additional suggestions, it would be of high value to verify the competition between ZTL-based ubiquitylation and UBP12/13 deubiquitylation in a heterologous system, although personally I believe that those experiments are beyond the scope of the current manuscript.

Reviewer #2 (Remarks to the Author):

In this interesting and nicely-executed study, Lee et al. identify a role for two deubiquitylases in the regulation of the plant circadian oscillator. This helps to explain how ZTL stability is controlled within the light/dark cycle in order to regulate the circadian oscillator. The authors demonstrate that GI is required for the deubiquitylases to become incorporated into a protein complex. This is not the first demonstration of that deubiquitylases affect the circadian oscillator, but has considerable novelty because it is the first demonstration of the mechanism by which deubiquitylases act within the plant circadian oscillator. Interestingly, deubiquitylases also act within the animal circadian oscillator, suggesting evolutionary conservation or convergent evolution of this mechanism.

Overall, I found the manuscript well written and in the most part convincing. In some places there seems to be some over-interpretation, and/or more nuance could be added to the descriptions. There are also some larger points below, that if tackled should improve the manuscript.

Major points

1. The authors find that UBP12/13 interact with GI but not with ZTL or ZTL targets. Is there any possibility of a false negative here on ZTL in the Y2H, given that ZTL will lack its chromophore when expressed in yeast so probably not be in its fully mature state?

2. The section around lines 106-117 discusses changes in period or amplitude on a number of occasions but it is difficult to read these data from the transcript abundance plots (Fig 2e-h). Presumably, the authors have performed some numerical analysis of the data to quantify period and amplitude in order to reach the conclusions. It would be very helpful to add these period and amplitude data to Fig. 2 to assist with its interpretation (I realize that such analysis from qRT-PCR can be noisy, but it does often work), potentially along with statistical analysis of the data if replication levels allow. It would make the descriptions of the results more compelling and allow proper statistical support of statements about changes in period, for example. The *biodare2* suite has some alternative algorithms from FFT-NLLS that might be useful here.

3. I find the statistical treatment of the data in Fig. 2i confusing and am not convinced it uses the most appropriate approach. The period differences between the genotypes seem quite small (e.g. between Col-0 and *ubp12-1*) and the data somewhat variable between replicates, so I am surprised that the statistical significance level for this comparison is the same as for Fig. 2a. The way that the statistical analysis is presented in Fig. 2i does not allow the reader to make all the comparisons that are required to interpret the data (e.g. is there a statistically-significant period difference between Col-0 and the complementation line?- presumably not, but this has to be inferred from the graph rather than being clear). Because Fig. 2a, c, i involve multiple comparisons, it would be more appropriate to analyse these data with some form of ANOVA and indicate statistical significance using different letter codes above each genotype to identify genotypes that are significantly different from each other. A final small point- why are there many fewer replicates for the activity-dead UBP12 in Fig. 2i (I wondered whether a proportion of the reps were arrhythmic, and so did not return a period value after FFT-NLLS analysis?); this needs some brief explanation.

4. I find the results for TOC1 protein abundance in the *ubp12/13* mutants (Fig. 3e) to be counterintuitive. If UBP12/13 activity helps to stabilize ZTL and GI, and mutation of these deubiquitylases causes very large decreases in the amount of ZTL protein (Fig. 3c), why is TOC1 protein abundance decreased given that loss of ZTL causes TOC1 over-accumulation (Mas et al. 2003)? I agree that these changes in TOC1 accumulation are reminiscent of those occurring in the *gi* mutant (Kim et al. 2007). Kim et al. PNAS 2011 show that partial reductions in ZTL protein abundance lead to increased TOC1 accumulation, so the authors' explanation that the small amount of remaining ZTL protein is important (Fig. 3c; line 189) does not seem aligned with the finding of Kim et al. that a decrease (rather than loss) of ZTL protein is associated with increased TOC1 protein abundance. If the authors feel this is a misunderstanding of the situation, please could they explain/interpret more explicitly the basis for the decrease in TOC1 protein abundance in the *ubp12/13* mutants, in order to reduce the potential for misunderstanding. Might UBP12/13 act upon other parts of the oscillator that have yet to be identified, giving unexpected results?

Minor points

1. The BiFC is nice, but I'm not completely convinced that a cytoplasmic signal is visible for UBP13-GI interaction (Fig. 1b). Unless this can be made clearer (e.g. by making the images larger?), it seems a good idea to add some nuance to the description around lines 70-74.

2. Lines 79-80; because the authors did not measure GI ubiquitination state, I suggest this is revised to "suggests" rather than "indicates that".

3. Lines 85-86; the authors did not measure protein stability / turnover within a timecourse, so some extra clarifying text should be added to justify why the authors interpret these data from one timepoint to indicate a change in ZTL protein stability (rather than ZTL protein synthesis, for example).

4. Lines 95-96; please can the authors be clear about whether they think that GI protein itself

forms the physical bridge from UBP12/13 to ZTL or whether product of GI activity forms might form the bridge.

5. Fig 2f; if UBP13 and GI are epistatic, why do the authors think that the period phenotype of *ubp13* is insensitive to the *gi* mutation? This seems to suggest some interesting differences between the behavior of *ubp12* and *ubp13* that could be expanded upon a little to add depth to the interpretation.

6. Lines 149-150; it is not true that GI-HA transcript abundance is 25% lower across the timecourse. Examination of the data would suggest that GI transcript abundance is only reduced substantially 8h after dawn, unless the difference is masked by the low transcript abundance values at other timepoints.

7. Line 155; "substantially" or "profoundly" might be a better word choice than "severely".

8. Line 184; "damped" is better than "dampened" (dampened also = to make wet).

9. Fig. 3a, c, e; the font size for the time axis is quite large and this means that all the text runs into itself (could it be made smaller- or clearer somehow?).

Reviewer #3 (Remarks to the Author):

The authors of this manuscript identify an association of the deubiquitylases, UBP12 and UBP13 (UBPs), with GIGANTEA (GI) as part of a complex with the F-box protein ZEITLUPE (ZTL). ZTL is a key component of the plant circadian system, as it controls the turnover of at least two essential clock transcriptional repressors, TOC1 and PRR5. Previously GI has been shown to act as a stabilizer of ZTL, through its role as a co-chaperone (with HSP90) of ZTL maturation. The Gendron lab has now shown through mass spectrometry of a decoy version of the ZTL protein complex, that these deubiquitylases likely associate with the GI/ZTL complex in vivo through their interaction with GI. They propose two roles for GI, one as a co-chaperone to ZTL (previously shown) and the other, their novel contribution, as a recruiter of UBPs which would also help stabilize ZTL.

Their findings add a new component to the GI/ZTL complex story, which helps to explain previous results which showed ZTL levels highly diminished in *gi* mutants. This is a significant addition to our growing appreciation of what comprises the GI/ZTL complex and how it functions. However, while the data are strongly suggestive of their model, there is little in the way of evidence of native protein interaction.

There is no direct in vivo validation of the UBPs interaction with the GI/ZTL complex at native protein levels. Interaction tests in Fig. 1 are via yeast two-hybrid or transient overexpression in protoplasts, both with GI. The primary evidence was from the IP MS, but there are numerous false positives likely and in fact UBP13 shows up in Group 4: Non-significant interactions with ZTL decoy in both Col-0 and *gi-2*. All other data presented to support their model are inferences based on genetics using *ubp* mutants in combination with other mutants and luciferase-based reporters. As well, there are additional very relevant questions that can now easily be addressed with the reagents available that would add value to this work and extend our understanding even more. Additional work needed includes:

a. There are UBP12/13 antibodies previously published (Cui et al. 2013), and/or UBP12/13-tag lines available from different publications (e.g. UBP12-RFP, UBP13-RFP; doi: 10.1038/nplants.2016.126) as well as their own (pUBP12/13::UBP12/13-YFP) that should be used to validate that these proteins co-exist with ZTL/GI in vivo via co-IP and testing with the ZTL ab

and/or crossed to GI:GI-HA and tested by co-IP.

Previous paper of Cui et al. 2013 suggested circadian oscillation of UPB protein in LL; this is relevant to the current work. Do the UBPs associate with GI differentially in a circadian manner or in a light/dark selective manner? This is what they assert in Fig S5: "At night, GI and UBP12/UBP13 dissociate from ZTL" The authors do not show any kind of time course under any kind of conditions to support this statement.

This could also be a significant factor in the circadian and/or L/D oscillation of ZTL. This can be addressed by GI/UBP co-IPs under native conditions over a time series. Alternatively, an estrogen-inducible UBP system has been published and could be used for a time series with equal induction at different circadian or light/dark times to test for differential association with GI.

b. The reduced levels of TOC1 in the ubp background could result from increased ubiquitylation of TOC1; are the UBPs there for ZTL or for the ZTL targets? or both? What is the ubiquitylation state of TOC1 (TMG) in the ubp mutants? The ubiquitylation state of ZTL? With such reduced levels of ZTL in the ubp background it is hard to believe that the lower TOC1 levels is due to some kind of super activity of this remaining ZTL, as they suggest: "the lack of repression of ZTL ligase activity by UBP12/UBP13 causes ZTL to actively ubiquitylate and degrade TOC1 before dusk and hinder TOC1 accumulation." If that were true then in the catalytically inactive UBP12C208S background, TOC1 levels should rise, since this UBP should still interact with the GI/ZTL complex and "repress ZTL activity".

What is the level of ZTL and GI in the UBP12C208S background?

c. The Fig S5 legend makes a number of unsupported assertions: "UBP12/UBP13 deubiquitylate and stabilize the GI-ZTL complex for proper accumulation of ZTL protein before dusk" , and "loss of UBP12 and UBP13 causes raised ubiquitylation of ZTL-GI complex" – do the authors mean that both ZTL and GI are deubiquitylated by the UBPs? Is GI deubiquitylated by the UBPs? They do not show evidence that the UBPs act on GI, though their figure suggest this.

Also stated: "UBP12/UBP13 also repress ZTL ligase activity toward the ZTL target, TOC1, leading to accumulation of TOC1 in the evening". There is no evidence that ZTL ligase activity toward TOC1 is affected by the UBPs. Maybe they are just regulating ZTL levels, not activity.

d. Some previous publications indicate that ztl mutants also result in diminished GI levels. Is it possible that a ZTL-GI association helps stabilize the UBP presence to co- deubiquitylate both GI and ZTL? Testing UPB co-IP with GI in the ztl mutant background would address this. Possibly transient co-expression tests in Arabidopsis ztl mutant protoplasts might be another way to address this. Or some kind of in vitro assay.

e. None of the known targets of ZTL show up in Table S1. Comment and compare to their earlier published results.

f. Line 85-87: these are separate transgenic lines – are protein differences just due different expression levels? check mRNA levels.

g. Fig 2E-F: why is CCA1 only measured? Significance of choosing that gene?

h. Fig 2C: "below linear range for quantification" ; should be "quantification".

i. Figure S2. Immunoprecipitation FLAG-His-ZTL decoy in the Col-0 or gi-2 genotypes. Should be "of"

We greatly appreciate the comments from the editor and reviewers. We were delighted to hear that the work is clearly presented and has significance for the circadian clock field, and we felt that the majority of the points raised were fair. In the letter below we summarize new experiments and text changes that were implemented. We attempt to address every concern that arose in the reviewers' remarks, and we feel that the changes improve the clarity and overall impact of the manuscript.

Reviewers' comments:

Reviewer #1 (Remarks to the Author):

Lee et al. provide a detailed and compelling study regarding the role of UBP12/13 in regulating the stability and function of ZTL in the circadian clock. The data presented address a major question in the field; namely, how does ZTL accumulate to high levels in the light without destabilizing and targeting its substrates for degradation? To address this question, they show that UBP12 and 13 are part of the ZTL-GI complex. More specifically, GI is required to bring UBP12/13 to the ZTL-GI complex where it can function as a deubiquitylase to stabilize both ZTL and its degradation targets. They demonstrate that the interaction is specifically through GI in both yeast-2-hybrid assays, with confirmation in vivo using IP-MS. They are able to further validate that UBP12/13 function in the same pathway as GI and that ZTL is epistatic to UBP12/13. The combined results provide a complete biochemical and genetic model of how ZTL targets proteins for degradation in a time of day specific manner, whereby ZTL activity is modulated by a ZTL-GI-UBP complex where UBP12/13 functions antagonistically to ZTL to create a balance between ubiquitylation and de-ubiquitylation of ZTL and its targets. In so doing they also resolve an oddity in the circadian clock community regarding how TOC1 levels are low when ZTL is destabilized in either GI or UBP12/13 mutants.

Although I cannot address technical aspects regarding the genetic approaches used in this manuscript (and leave that for other reviewers), the biochemical and molecular approaches outlined in this manuscript are well conducted and compelling. Overall the manuscript is well written and is of high-significance to the field. For these reasons I support publication of the manuscript in Nature Communications as is.

If forced to make some additional suggestions, it would be of high value to verify the competition between ZTL-based ubiquitylation and UBP12/13 deubiquitylation in a heterologous system, although personally I believe that those experiments are beyond the scope of the current manuscript.

Response: This is an excellent suggestion. We routinely use mammalian tissue culture cells to test the roles of plant E3 ubiquitin ligases in ubiquitylating their target proteins. We have been able to reconstitute ZTL targeting of CHE (Lee and Feke et al. 2018) and TOC1 (data not published yet). Others have also shown that GI can associate with the FKF1 (a ZTL homolog) LOV domain in a light-dependent manner in mammalian tissue culture cells. We made multiple attempts to reconstitute the ZTL/GI/UBP complex while co-expressing TOC1 in mammalian cells. This would allow us to track ubiquitylation of TOC1 in the light and dark and in the presence or absence of UBP12 (as the reviewer suggested above). Unfortunately, we faced technical challenges and were not able to co-express all four components together. We tried altering transfection parameters and also blocking proteasome function but were not able to achieve sufficient co-expression. Most likely we will need to perform additional optimization of the transfection conditions or acquire a new cell type to get all four proteins expressed at the same time.

Reviewer #2 (Remarks to the Author):

In this interesting and nicely-executed study, Lee et al. identify a role for two deubiquitylases in the regulation of the plant circadian oscillator. This helps to explain how ZTL stability is controlled within the light/dark cycle in order to regulate the circadian oscillator. The authors demonstrate that GI is required for the deubiquitylases to become incorporated into a protein complex. This is not the first demonstration of that deubiquitylases affect the circadian oscillator, but has considerable novelty because it is the first demonstration of the mechanism by which deubiquitylases act within the plant circadian oscillator. Interestingly, deubiquitylases also act within the animal circadian oscillator, suggesting evolutionary conservation or convergent evolution of this mechanism.

Overall, I found the manuscript well written and in the most part convincing. In some places there seems to be some over-interpretation, and/or more nuance could be added to the descriptions. There are also some larger points below, that if tackled should improve the manuscript.

Major points

1. The authors find that UBP12/13 interact with GI but not with ZTL or ZTL targets. Is there any possibility of a false negative here on ZTL in the Y2H, given that ZTL will lack its chromophore when expressed in yeast so probably not be in its fully mature state?

Response: Yeast two-hybrid has been a successful strategy for studying protein-protein interactions with the ZTL LOV domain (Mas et al. 2003, Lee and Feke et al. 2018 for example). The question of chromophore presence is an interesting one, and thus we searched for references showing that Flavin mononucleotide, the ZTL chromophore, is present in *S. cerevisiae*. There were multiple references showing that yeast do make FMN and FAD, and one that we found showing reporters in the cytosolic space can utilize free FMN (i.e. Pallotta et al. 1998 *FEBS letters* and Tielker et al. 2009 *Eukaryotic Cell*). We hope that this will alleviate concerns that we encountered a false negative yeast two-hybrid result because the FMN isn't present for association with the LOV domain of ZTL. Furthermore, we lose the interaction between ZTL and the UBPs in the absence of GI suggesting that GI is necessary for the interaction.

2. The section around lines 106-117 discusses changes in period or amplitude on a number of occasions but it is difficult to read these data from the transcript abundance plots (Fig 2e-h). Presumably, the authors have performed some numerical analysis of the data to quantify period and amplitude in order to reach the conclusions. It would be very helpful to add these period and amplitude data to Fig. 2 to assist with its interpretation (I realize that such analysis from qRT-PCR can be noisy, but it does often work), potentially along with statistical analysis of the data if replication levels allow. It would make the descriptions of the results more compelling and allow proper statistical support of statements about changes in period, for example. The biodare2 suite has some alternative algorithms from FFT-NLLS that might be useful here.

Response: We entered our data into Biodare2 to make period, phase, and amplitude estimates for the qPCR data that we generated. Because our data has lower resolution than recommended for Biodare2 (we used 3 or 4 hour resolution over two days) we analyzed the data using 4 algorithms, FFT-NLLS, MESA, LS Periodogram, and Spectrum Resampling. We then compared the period calls from the four analyses to published period calls and our empirical data from figure 2a-d. None of the algorithms were

able to call the *ubp13-1* period from figure 2h (~1 hour short in our hands), but LS Periodogram was able to call all other mutant and wild-type periods correctly relative to all previous studies, thus we chose to present the LS Periodogram data for the experiments. We added this data as Supplementary Table S3. We added new text to discuss the results of Table S3. Briefly, our original conclusions were supported by the analysis but we were also able to present a more refined discussion of the genetic data. The new text can be found in lines 121-145. We appreciate this suggestion.

3. I find the statistical treatment of the data in Fig. 2i confusing and am not convinced it uses the most appropriate approach. The period differences between the genotypes seem quite small (e.g. between Col-0 and *ubp12-1*) and the data somewhat variable between replicates, so I am surprised that the statistical significance level for this comparison is the same as for Fig. 2a. The way that the statistical analysis is presented in Fig. 2i does not allow the reader to make all the comparisons that are required to interpret the data (e.g. is there a statistically-significant period difference between Col-0 and the complementation line?- presumably not, but this has to be inferred from the graph rather than being clear). Because Fig. 2a, c, i involve multiple comparisons, it would be more appropriate to analyse these data with some form of ANOVA and indicate statistical significance using different letter codes above each genotype to identify genotypes that are significantly different from each other.

Response: Thank you for this excellent point. In response to this concern we rethought our statistical analysis approach and agree that it is not appropriate for this experiment type. In this experiment we are analyzing a population of T1 transgenic lines to determine whether the UBP12C208S mutant can rescue the clock defects of the *ubp12-1* mutant to determine if the deubiquitylating activity of UBP12 is necessary for its clock function. In response to the above concern, we decided to exclude the statistical analysis and rather took a qualitative approach and counted the number of lines that rescue the mutant phenotype. We defined “rescue” as lines having a period longer than one standard deviation above the average of the *ubp12-1* mutant. Strikingly, the UBP12C208S transgenic population only had one line with period longer than *ubp12-1* while 1/3 of the wild-type UBP12 rescue lines were longer than the *ubp12-1* mutant. Because there is experiment-to-experiment variation even the *ubp12-1* has some plants (13%) that fall outside its own mean and standard deviation and the wild type has ~62% that fall outside of the *ubp12-1* average. This qualitative approach shows that the wild-type rescue construct has the potential to rescue the clock defect while the UBP12C208S does not. This is shown in figure 2i and 2j. We also added new text describing this analysis in lines 156 to 163. As noted below we increased the number of replicates of the experiment.

A final small point- why are there many fewer replicates for the activity-dead UBP12 in Fig. 2i (I wondered whether a proportion of the reps were arrhythmic, and so did not return a period value after FFT-NLLS analysis?); this needs some brief explanation.

Response: We checked this and all of the lines we were able to identify were rhythmic. For an unknown reason we identified fewer transgenics for the protease-dead UBP12CS. In response, we have included new data for the experiment. The number of lines is clearly noted in figure 2j (N=20 now for the UBP12CS transgenic lines) as well as the percentage of lines that pass our threshold for rescuing the *ubp12-1* short period phenotype (as described above).

4. I find the results for TOC1 protein abundance in the *ubp12/13* mutants (Fig. 3e) to be counterintuitive. If UBP12/13 activity helps to stabilize ZTL and GI, and mutation of these deubiquitylases causes very large decreases in the amount of ZTL protein (Fig. 3c), why is TOC1 protein

abundance decreased given that loss of ZTL causes TOC1 over-accumulation (Mas et al. 2003)? I agree that these changes in TOC1 accumulation are reminiscent of those occurring in the *gi* mutant (Kim et al. 2007). Kim et al. PNAS 2011 show that partial reductions in ZTL protein abundance lead to increased TOC1 accumulation, so the authors' explanation that the small amount of remaining ZTL protein is important (Fig. 3c; line 189) does not seem aligned with the finding of Kim et al. that a decrease (rather than loss) of ZTL protein is associated with increased TOC1 protein abundance. If the authors feel this is a misunderstanding of the situation, please could they explain/interpret more explicitly the basis for the decrease in TOC1 protein abundance in the *ubp12/13* mutants, in order to reduce the potential for misunderstanding. Might UBP12/13 act upon other parts of the oscillator that have yet to be identified, giving unexpected results?

Response: This is an excellent observation, and we agree that it is counterintuitive that TOC1 levels are decreased in the *ubp* (or *gi-2*) mutants when ZTL levels are also low. While we do not want to speculate too wildly, we think that in the *ubp* mutants that the remaining ZTL is likely still able to engage with GI and HSP90 which help it fold into an active F-box protein that can mediate ubiquitylation of target proteins. But without UBP at the GI/ZTL complex, ZTL inappropriately ubiquitylates its targets during the day and then is degraded by an unknown mechanism, possibly autoubiquitylation. In the PNAS paper from Kim *et al.* in 2011 they inhibit the function of HSP90 which 1) lowers ZTL levels but 2) also prevents maturation of the protein likely making it less functional in the SCF complex due to lack of proper folding. This may explain why the target proteins increase when HSP90 is inhibited. We chose not to speculate on this to a great extent without empirical proof, but we have added text in lines 217-223 to make this point more overt in the manuscript.

Minor points

1. The BiFC is nice, but I'm not completely convinced that a cytoplasmic signal is visible for UBP13-GI interaction (Fig. 1b). Unless this can be made clearer (e.g. by making the images larger?), it seems a good idea to add some nuance to the description around lines 70-74.

Response: Thank you for the suggestion. We enlarged the figures to try to increase the visibility of nuclear signal. We also modified the text to read "our BiFC results show that UBP12 and UBP13 interact with GI in both compartments with strong signal in the nucleus and weaker but detectable signal in the cytoplasm". This is at line 72 to 77.

2. Lines 79-80; because the authors did not measure GI ubiquitination state, I suggest this is revised to "suggests" rather than "indicates that".

Response: This change was made.

3. Lines 85-86; the authors did not measure protein stability / turnover within a timecourse, so some extra clarifying text should be added to justify why the authors interpret these data from one timepoint to indicate a change in ZTL protein stability (rather than ZTL protein synthesis, for example).

Response: We agree that our conclusion is not supported by the data as was also mentioned by another reviewer. Because this has minor importance to our overall conclusions we decided to remove it for clarity.

4. Lines 95-96; please can the authors be clear about whether they think that GI protein itself forms the physical bridge from UBP12/13 to ZTL or whether product of GI activity forms might form the bridge.

Response: We changed the text to say that the GI protein physically bridges the interaction between ZTL and the UBPs.

5. Fig 2f; if UBP13 and GI are epistatic, why do the authors think that the period phenotype of *ubp13* is insensitive to the *gi* mutation? This seems to suggest some interesting differences between the behavior of *ubp12* and *ubp13* that could be expanded upon a little to add depth to the interpretation.

Response: This is an excellent observation. We find that UBP12 and UBP13 are not completely redundant. UBP13 seems to play a slightly different role in the clock than UBP12. We have added new statistical analyses using Biodare2 and additional text in lines 121-145 discussing the potential differences.

6. Lines 149-150; it is not true that GI-HA transcript abundance is 25% lower across the timecourse. Examination of the data would suggest that GI transcript abundance is only reduced substantially 8h after dawn, unless the difference is masked by the low transcript abundance values at other timepoints.

Response: This is true. We changed the text to read that the GI-HA transcript abundance is approximately 25% lower at the peak of expression (ZT8).

7. Line 155; “substantially” or “profoundly” might be a better word choice than “severely”.

Response: We changed the text to “substantially”. Thank you for this suggestion.

8. Line 184; “damped” is better than “dampened” (dampened also = to make wet).

Response: We changed to “damped”.

9. Fig. 3a, c, e; the font size for the time axis is quite large and this means that all the text runs into itself (could it be made smaller- or clearer somehow?).

Response: We increased the spacing between the numbers.

Reviewer #3 (Remarks to the Author):

The authors of this manuscript identify an association of the deubiquitylases, UBP12 and UBP13 (UBPs), with GIGANTEA (GI) as part of a complex with the F-box protein ZEITLUPE (ZTL). ZTL is a key component of the plant circadian system, as it controls the turnover of at least two essential clock transcriptional repressors, TOC1 and PRR5. Previously GI has been shown to act as a stabilizer of ZTL, through its role as a co-chaperone (with HSP90) of ZTL maturation. The Gendron lab has now shown through mass spectrometry of a decoy version of the ZTL protein complex, that these deubiquitylases likely associate with the GI/ZTL complex in vivo through their interaction with GI. They propose two roles for GI, one as a co-chaperone to ZTL (previously shown) and the other, their novel contribution, as a recruiter of UBPs which would also help stabilize ZTL.

Their findings add a new component to the GI/ZTL complex story, which helps to explain previous results

which showed ZTL levels highly diminished in *gi* mutants. This is a significant addition to our growing appreciation of what comprises the GI/ZTL complex and how it functions. However, while the data are strongly suggestive of their model, there is little in the way of evidence of native protein interaction.

There is no direct *in vivo* validation of the UBPs interaction with the GI/ZTL complex at native protein levels. Interaction tests in Fig. 1 are via yeast two-hybrid or transient overexpression in protoplasts, both with GI. The primary evidence was from the IP MS, but there are numerous false positives likely and in fact UBP13 shows up in Group 4: Non-significant interactions with ZTL decoy in both Col-0 and *gi-2*. All other data presented to support their model are inferences based on genetics using *ubp* mutants in combination with other mutants and luciferase-based reporters. As well, there are additional very relevant questions that can now easily be addressed with the reagents available that would add value to this work and extend our understanding even more. Additional work needed includes:

a. There are UBP12/13 antibodies previously published (Cui *et al.* 2013), and/or UBP12/13-tag lines available from different publications (e.g. UBP12-RFP, UBP13-RFP; doi: 10.1038/nplants.2016.126) as well as their own (pUBP12/13::UBP12/13-YFP) that should be used to validate that these proteins co-exist with ZTL/GI *in vivo* via co-IP and testing with the ZTL ab and/or crossed to GI:GI-HA and tested by co-IP.

Response: Thank you for this excellent suggestion. This was the main focus of our experimental efforts during the revision period. We made multiple attempts to perform the exact requested experiments but were prevented from doing so due to technical constraints (outlined below). Fortunately, we were able to reconstitute the UBP/GI/ZTL complex in *Nicotiana* leaves using transient expression (new Fig.1e). We show that the interaction between ZTL and the UBPs is dependent on GI, confirming the results of the IP-MS experiment that we present in Fig. 1d and the complementary results from figures 1a-c.

Additionally, during the revision period the Millar group published an extensive GI protein interaction study that serves as an ideal complementary study to our own. Initially, we showed that the ZTL decoy co-immunoprecipitates with GI, UBP12, and UBP13 (Lee and Feke *et al.* 2018). Subsequently, the Millar group performed a time-course immunoprecipitation followed by mass spectrometry study with constitutively expressed GI that shows that GI interacts with UBP12, UBP13, and ZTL, partially fulfilling the requested experiment (Krahmer *et al.* 2018, FEBS Letters). We believe their work bolsters our conclusions by showing that a separate group using different techniques can corroborate the studies that we have performed.

We have included new discussion (lines 99-109) describing the transient co-IP experiment and new discussion (lines 63-65) of the recent FEBS Letters manuscript from the Millar lab (Krahmer *et al.* 2017) to reflect these findings.

These were the two additional experiments that we attempted during the revision period but that failed for technical reasons.

1. We requested and received the antibody from the Cui group, but the shipping of the antibody from China to the US was difficult. It was held at Customs for months, then returned to the Cui group and freeze-dried, and then subsequently shipped to us. Unsurprisingly, it failed to detect any protein bands from total *Arabidopsis* protein extract. We conferred with the Cui group to ensure that our protocol was correct for detecting UBP12 and UBP13, suggesting that the reagent was damaged in transport. This severely limited the approaches we could take to study the ZTL/GI/UBP interaction.

2. We next attempted to cross the *GI::GI:HA* line to the *pUBP12:UBP12-YFP* line from our studies and the transgenes were silenced. We also propagated the *pUBP12:UBP12-YFP* to a homozygous generation but it also was silenced, possibly due to the presence of the *CCA1_{promoter}::Luciferase* transgenic background.

We hope that the combination of experiments that we present (new Fig.1e), along with the work from the Millar group give confidence that our model for the UBPGI/ZTL complex is correct according to our current knowledge.

Previous paper of Cui et al. 2013 suggested circadian oscillation of UPB protein in LL; this is relevant to the current work. Do the UBPs associate with GI differentially in a circadian manner or in a light/dark selective manner? This is what they assert in Fig S5: "At night, GI and UB12/UB13 dissociate from ZTL" The authors do not show any kind of time course under any kind of conditions to support this statement. This could also be a significant factor in the circadian and/or L/D oscillation of ZTL. This can be addressed by GI/UBP co-IPs under native conditions over a time series. Alternatively, an estrogen-inducible UB system has been published and could be used for a time series with equal induction at different circadian or light/dark times to test for differential association with GI.

Response: As mentioned above, this data was recently published in FEBS Letters in which the authors perform a time-course IP-MS with the GI protein (Krahmer *et al.* 2017) Briefly, it shows that GI and the UBPs associate throughout the day and night and that no dissociation is observed. Thank you for this suggestion. We have added text describing the results in the manuscript in lines 63-65.

b. The reduced levels of TOC1 in the *ubp* background could result from increased ubiquitylation of TOC1; are the UBPs there for ZTL or for the ZTL targets? or both? What is the ubiquitylation state of TOC1 (TMG) in the *ubp* mutants? The ubiquitylation state of ZTL? With such reduced levels of ZTL in the *ubp* background it is hard to believe that the lower TOC1 levels is due to some kind of super activity of this remaining ZTL, as they suggest: "the lack of repression of ZTL ligase activity by UB12/UB13 causes ZTL to actively ubiquitylate and degrade TOC1 before dusk and hinder TOC1 accumulation."

Response: To our knowledge, there is no example in the literature showing ubiquitylation of TOC1 or ZTL *in planta*. It is likely that this is due to technical challenges in detecting ubiquitylated, and thus highly unstable, forms of these proteins. The field standard has been to look at protein levels, which we did in figure 3. The technical difficulty is increased in this case because ZTL and TOC1 protein levels in the *ubp* mutants start lower than wild type making it less likely that we would be able to capture ubiquitylated forms of the protein.

Despite this, we tried two experiments. First, we attempted to detect ubiquitylated forms of ZTL or TOC1 by looking for higher molecular weight forms of the proteins on our western blots. We looked at long exposure blots and modified the brightness and contrast, but no bands were visible before overexposure of the blot. Second, we tried to reconstitute the ZTL/GI/UBP/TOC1 complex in mammalian tissue culture cells. In our opinion this is the best experiment to answer the above questions about ubiquitylation states because it is done in a heterologous system in the absence of other plant proteins that could confound the interpretation of the results. Furthermore, we were able to show ubiquitylation of a ZTL target by ZTL using this system previously. In this case we were unable to express all four of the proteins to sufficient levels to perform the ubiquitylation assays. This was discussed in more detail in regards to a suggestion from Reviewer 1.

Because we could not complete these experiments we opted to alter the figure legend of Figure S5 to reduce any speculative statements.

If that were true then in the catalytically inactive UBP12C208S background, TOC1 levels should rise, since this UBP should still interact with the GI/ZTL complex and "repress ZTL activity".
What is the level of ZTL and GI in the UBP12C208S background?

Response: We hypothesize that the UBP12C208S would not affect the level of TOC1 if expressed in the *ubp12-1* mutant background because it has no protease activity and can't function as a deubiquitylase and cleave ubiquitin from targets (Cui *et al.* 2013), and it does not rescue the *ubp12-1* mutant (Fig.2i-j). We think that UBP12 or UBP13 could repress ZTL activity in wild-type form by acting as a deubiquitylase but could not perform this function when catalytically dead. We apologize if this was not clear.

c. The Fig S5 legend makes a number of unsupported assertions: "UBP12/UBP13 deubiquitylate and stabilize the GI-ZTL complex for proper accumulation of ZTL protein before dusk", and "loss of UBP12 and UBP13 causes raised ubiquitylation of ZTL-GI complex" – do the authors mean that both ZTL and GI are deubiquitylated by the UBPs? Is GI deubiquitylated by the UBPs? They do not show evidence that the UBPs act on GI, though their figure suggest this.

Also stated: "UBP12/UBP13 also repress ZTL ligase activity toward the ZTL target, TOC1, leading to accumulation of TOC1 in the evening". There is no evidence that ZTL ligase activity toward TOC1 is affected by the UBPs. Maybe they are just regulating ZTL levels, not activity.

Response: Thank you for these suggestions. We rewrote the figure legend to increase clarity and accuracy.

d. Some previous publications indicate that *ztl* mutants also result in diminished GI levels. Is it possible that a ZTL-GI association helps stabilize the UBP presence to co- deubiquitylate both GI and ZTL? Testing UPB co-IP with GI in the *ztl* mutant background would address this. Possibly transient co-expression tests in Arabidopsis *ztl* mutant protoplasts might be another way to address this. Or some kind of in vitro assay.

Response: This is an excellent idea and is something that could be investigated in the future with some of the abovementioned tools. Our yeast two-hybrid results suggest that ZTL is not necessary for UBP to interact with GI, and GI has numerous ZTL-independent roles. It would be quite interesting to investigate whether UBP12 or UBP13 regulate those ZTL-independent functions of GI and whether ZTL has some direct or indirect role in regulating GI. We intentionally focused our work on the role of the UBPs in regulating ZTL in order to answer a very focused question. We feel that studying the roles of ZTL, UBP12, and UBP13 on GI activity would significantly widen the scope of the work beyond what we intended to show in this manuscript.

e. None of the known targets of ZTL show up in Table S1. Comment and compare to their earlier published results.

Response: For this study we chose the ZT9 time point because we wanted to enrich for the ZTL/GI/UBP complex. At this time point the known targets of ZTL are not present. This matches the results that we showed previously where we detected the targets at later time points (Lee and Feke *et al.* 2018).

f. Line 85-87: these are separate transgenic lines – are protein differences just due different expression levels? check mRNA levels.

Response: This was also mentioned by reviewer 1. We agree that our conclusion is not supported by the data. Because this has minor importance to our overall conclusion we decided to remove it for clarity.

g. Fig 2E-F: why is CCA1 only measured? Significance of choosing that gene?

Response: There are three reasons why we chose *CCA1* for this experiment: 1) *CCA1* is a central component of the core *Arabidopsis* circadian clock, and its expression pattern directly reports on clock activity, 2) *CCA1* is widely used in circadian clock studies for qRT-PCR because of its robust rhythmicity, and 3) *CCA1* is a direct transcriptional target of TOC1 making it a useful reporter for studies of the GI/ZTL complex which regulates the stability of TOC1 and thus periodicity, phase, and amplitude of *CCA1* mRNA expression.

h. Fig 2C: “below linear range for quantification” ; should be “quantification”.

Response: Thank you. We fixed this in the manuscript.

i. Figure S2. Immunoprecipitation FLAG-His-ZTL decoy in the Col-0 or gi-2 genotypes. Should be “of”

Response: Thank you. We fixed this in the manuscript.

Reviewers' comments:

Reviewer #2 (Remarks to the Author):

The authors have performed a fairly extensive revision of their manuscript, with new data, analysis and interpretation. They should be commended for taking these steps to improve the paper considerably.

In my previous review, the appropriateness of the statistical treatment of some data in Fig. 2 was queried. The authors have taken an innovative alternative approach to this- recognizing that the data are probably noisy- and Fig. 2j presents a very satisfactory solution to this issue. If the data or between-replicate responses are variable, the authors should consider it OK to mention that in the manuscript (I feel in general that authors can be reluctant to mention when data are variable in this way, even though it's pretty normal in biology). I think the authors handled that very well in the section around lines 140-147.

Overall, this is an excellent study and I hope the journal considers it a candidate for publication. My comments here are mainly to serve as a guide for making the manuscript as good as possible.

Larger points

1. The authors might want to check the correct paper structure for this journal. I recall the format used has an Introduction section. Likewise, the section from line 206 onwards appears to be Discussion and could (potentially) be headed as such. At the moment the structure is quite monolithic.

2. Examining the timecourse plots in in Fig. 2, I noticed that one point of each timecourse in each panel does not seem to have any error bars. I suspect this is a result of how the authors normalized the data, because this point appears to have the value of 1.0 in each timecourse. For absolute clarity, please could the authors add some explanation of this to the figure legend, along with any relevant information about which timepoint was chosen as this normalizer and why?

3. The authors might want to consider making Fig. S5 a main Fig. 4. It can sometimes help to show these types of cartoons more upfront in order to broaden the scope/impact of the paper. It can especially help those outside the field, new students to the field, etc.

Minor points

Line 9, is it correct to say that plants "constantly" survey their light conditions when the light input to the clock might be gated? This could be worded in a slightly more nuanced way to take that into account.

Line 17, is it really the case that UBP12 and UBP13 have completely opposite "biochemical functions" to ZTL? An opposite biochemical function to ZTL might be to emit light, for example. Perhaps this could be reworded because I think the authors are referring specifically to ubiquitination biochemistry.

Lines 35-40, In the introduction section that starts to focus upon more detailed aspects of oscillator function (e.g. 35-40), it might be worth prefixing a sentence with, "In Arabidopsis," because this would be most accurate.

Line 141, and elsewhere (please check throughout) should be "These data suggest" etc. because "data" is plural.

Line 203, "lowered levels of the TOC1 protein result in shortened period" – does this need a literature reference?

Line 222, rephrase to -> "counterbalancing the role of ZTL in..." [currently sloppy scientific English]

Line 230, are clocks "designed"? This reviewer tends to take the position that they evolved.

Fig. 2 panels and legend- possibly worth changing "Hours in LL" to "Time in constant light (h)" or similar, because LL is clock community jargon and this is a general interest journal. I have, for example, seen some readers interpret "LL" incorrectly as "low light" even when the abbreviation is obvious within the legend.

Fig. S1 legend- "protein structure" might be refined to "protein domain structure" because this isn't a 3D structure.

Reviewer #3 (Remarks to the Author):

In this resubmission the authors attempted to address the concerns of this reviewer with respect to in vivo validation of UBP interaction with GI/ZTL at native levels of expression. Attempts to use anti-UBP antibodies or a tagged UBP line were unsuccessful due either to antibody dysfunction or transgene silencing after crossing. They rely instead on a recent publication from the Millar lab where a line constitutively expressing GI was used for IP-mass spec and recovered UBP proteins and ZTL, independently showing that GI associates with UBPs and ZTL, though it is not shown whether GI interaction with UPB is related to GI interaction with ZTL.

They also rely on the one new experiment included which is transient overexpression in *N. benthamiana* of GI, ZTL and UBP proteins in various combinations to support that GI is a necessary intermediary for the co-IP of UBP and ZTL. This is the only direct evidence in the publication for a trimeric complex of the 3, as depicted in Fig 1f, allowing the caption of Fig.1 to be stated. It is good that this is now shown.

The authors cannot demonstrate that ZTL or TOC1 or any other component is ubiquitinated, but rely on the failure of the pUBP12::UBP12CS-YFP lines to rescue the ubp mutants as evidence that deubiquitinating functions of UBP12 are necessary for the clock [in addition to the lower protein levels of ZTL and GI in Fig. 3]. Has it been shown that this mutation does not destabilize the protein? It should be established that this mutant protein is expressed at levels comparable to the WT UBP12 lines that did rescue. This should be a relatively simple experiment and would establish that the activity of the UBPs, not just their presence, is necessary for the trimeric complex function.

Otherwise, the other comments submitted by the authors in response are acceptable.

Minor:

Legend S5:

"At night" would be more accurate to say "In the dark".

Reviewers' comments:

Reviewer #2 (Remarks to the Author):

The authors have performed a fairly extensive revision of their manuscript, with new data, analysis and interpretation. They should be commended for taking these steps to improve the paper considerably.

In my previous review, the appropriateness of the statistical treatment of some data in Fig. 2 was queried. The authors have taken an innovative alternative approach to this- recognizing that the data are probably noisy- and Fig. 2j presents a very satisfactory solution to this issue. If the data or between-replicate responses are variable, the authors should consider it OK to mention that in the manuscript (I feel in general that authors can be reluctant to mention when data are variable in this way, even though it's pretty normal in biology). I think the authors handled that very well in the section around lines 140-147.

Overall, this is an excellent study and I hope the journal considers it a candidate for publication. My comments here are mainly to serve as a guide for making the manuscript as good as possible.

Larger points

1. The authors might want to check the correct paper structure for this journal. I recall the format used has an Introduction section. Likewise, the section from line 206 onwards appears to be Discussion and could (potentially) be headed as such. At the moment the structure is quite monolithic.

Response: Thank you for this comment. This manuscript was originally a direct transfer from another Nature journal with a different format. We have now broken the manuscript into the subsections per Nature Communications format. This required minor additions of text to the end of the Introduction and beginning of the Results section (Lines 75-86).

2. Examining the timecourse plots in in Fig. 2, I noticed that one point of each timecourse in each panel does not seem to have any error bars. I suspect this is a result of how the authors normalized the data, because this point appears to have the value of 1.0 in each timecourse. For absolute clarity, please could the authors add some explanation of this to the figure legend, along with any relevant information about which timepoint was chosen as this normalizer and why?

Response: Thank you for this comment. We did not normalize the data to any individual time point. The absence of the error bars was due to a quirk of the graphing software (GraphPad Prism) that we used. This did not affect our statistical analyses since we used the raw data. We fixed this issue and the error bars are now present. Again, thank you for your careful consideration and helping us catch this error.

3. The authors might want to consider making Fig. S5 a main Fig. 4. It can sometimes help to show these types of cartoons more upfront in order to broaden the scope/impact of the paper. It can especially help those outside the field, new students to the field, etc.

Response: We are happy to do this! Figure S5 is now Figure 4. We have changed the figure callouts to reflect this change.

Minor points

Line 9, is it correct to say that plants “constantly” survey their light conditions when the light input to the clock might be gated? This could be worded in a slightly more nuanced way to take that into account.

Response: Agreed. We changed “constantly” to “must”.

Line 17, is it really the case that UBP12 and UBP13 have completely opposite “biochemical functions” to ZTL? An opposite biochemical function to ZTL might be to emit light, for example. Perhaps this could be reworded because I think the authors are referring specifically to ubiquitination biochemistry.

Response: We changed this to read “which regulate clock period and protein ubiquitylation in a manner opposite to ZTL”.

Lines 35-40, In the introduction section that starts to focus upon more detailed aspects of oscillator function (e.g. 35-40), it might be worth prefixing a sentence with, “In Arabidopsis,” because this would be most accurate.

Response: We changed this to read “One way that Arabidopsis senses the end of the day”.

Line 141, and elsewhere (please check throughout) should be “These data suggest” etc. because “data” is plural.

Response: Thank you. We believe we have corrected all errors.

Line 203, “lowered levels of the TOC1 protein result in shortened period” – does this need a literature reference?

Response: For accuracy we changed this to read “This is similar to the effects of the gi-2 mutant, where TOC1 protein levels never accumulate to full wild-type levels (Kim et al., 2007). This suggests that the period effects of the ubp12 and ubp13 mutants may be caused by the same mechanism as the short period of the gi-2 mutant.”

Line 222, rephrase to -> “counterbalancing the role of ZTL in...” [currently sloppy scientific English]

Response: Change made as suggested.

Line 230, are clocks “designed”? This reviewer tends to take the position that they evolved.

Response: Agreed and removed.

Fig. 2 panels and legend- possibly worth changing “Hours in LL” to “Time in constant light (h)” or similar, because LL is clock community jargon and this is a general interest journal. I have, for example, seen some readers interpret “LL” incorrectly as “low light” even when the abbreviation is obvious within the legend.

Response: We changed all occurrences in the text and figures to read “constant light” rather than “LL”. We agree this helps improve readability for all audiences.

Fig. S1 legend- “protein structure” might be refined to “protein domain structure” because this isn’t a 3D structure.

Response: Agreed and changed.

Reviewer #3 (Remarks to the Author):

In this resubmission the authors attempted to address the concerns of this reviewer with respect to in vivo validation of UBP interaction with GI/ZTL at native levels of expression. Attempts to use anti-UBP antibodies or a tagged UBP line were unsuccessful due either to antibody dysfunction or transgene silencing after crossing. They rely instead on a recent publication from the Millar lab where a line constitutively expressing GI was used for IP-mass spec and recovered UBP proteins and ZTL, independently showing that GI associates with UBPs and ZTL, though it is not shown whether GI interaction with UPB is related to GI interaction with ZTL.

They also rely on the one new experiment included which is transient overexpression in *N. benthamiana* of GI, ZTL and UBP proteins in various combinations to support that GI is a necessary intermediary for the co-IP of UBP and ZTL. This is the only direct evidence in the publication for a trimeric complex of the 3, as depicted in Fig 1f, allowing the caption of Fig.1 to be stated. It is good that this is now shown.

The authors cannot demonstrate that ZTL or TOC1 or any other component is ubiquitinated, but rely on the failure of the pUBP12::UBP12CS-YFP lines to rescue the ubp mutants as evidence that deubiquitylating functions of UBP12 are necessary for the clock [in addition to the lower protein levels of ZTL and GI in Fig. 3]. Has it been shown that this mutation does not destabilize the protein? It should be established that this mutant protein is expressed at levels comparable to the WT UBP12 lines that did rescue. This should be a relatively simple experiment and would establish that the activity of the UBPs, not just their presence, is necessary for the trimeric complex function.

Response: This is an interesting and important idea that we had not considered previously. The suggested experiment is more difficult than described for technical reasons, but we have provided data from another experiment that we performed in hopes that it will suffice. The reason that we felt the suggested experiment is unfeasible are listed here:

1. As stated in our previous “response to reviewers” the *UBP12-GFP* transgenes have been problematic because they are silenced after the T1 generation (possibly due to the presence of the *CCA1_{prom}::Luciferase* marker background), which previously prevented us from completing time course co-IPs with the material. Unfortunately, this would then require that we test the protein levels in the T1 lines.
2. To test the protein levels in the T1 generation, we would need to repeat the experiment to generate new traces, select multiple T1 plants for protein measurements in order to avoid stochastic differences in protein expression, and include mRNA measurements to benchmark the protein levels. Further complicating this, the UBP protein levels cycle across the day (Cui *et al.*, *Plant Physiology* 2013 Figure 5F). Presumably, the complemented and non-complemented lines would show differences in UBP protein level due to shifts in phase regardless of changes in the absolute protein level. Ideally, we would collect a time course and calculate the amplitude of protein expression (similar to what we did for TOC1, ZTL and GI in our manuscript). This is not ideal or even feasible with T1 plants. The difficulty is also

enhanced because we would need to attempt this with multiple T1 plants to prevent bias arising from analyzing a single T1 line.

Despite these issues, we strongly agree with the reviewer's idea, and we think that determining whether the UBP12^{C208S} mutant protein is less stable than the wild type UBP12 protein is important. Thus, we devised a different strategy that avoids the pitfalls mentioned above. We transiently transfected GFP-tagged wild type and C208S variants of UBP12 into tobacco. It was shown previously that Arabidopsis UBP12 can function in tobacco leaves in place of the tobacco UBP12 (Ewan *et al. New Phytologist* 2011 Figure 5) making this a good transient system for these types of assays. This paper also included western blots showing that the UBP12^{C208S} mutant protein can accumulate, but no comparison or quantification was provided.

We imaged the GFP signal from transfected cells at the same setting from both wild type and C208S transfections and then quantified GFP signal intensity. We chose to quantify the nuclear signal because the nucleus is easily defined using the AS2-mCherry, and the nuclear signal is uniform. To avoid any bias we quantified two distinct areas of each nuclei and avoided the nucleolus and nuclear periphery.

In this experiment we saw no difference in signal intensity between the UBP12 wild type and C208S mutant suggesting that the mutation does not affect the accumulation of the UBP12 protein. This is included in a new version of figure S4, the legend for figure S4, and in text lines 181-184. Figure S4 includes two representative images for each protein combination, a diagram of the strategy for quantification, and the quantification results.

While this is not the exact experiment that was requested, we feel that it fulfills the "spirit" of the request and are also happy that it significantly strengthens the manuscript as intended. We thank the reviewer for this thoughtful comment.

Otherwise, the other comments submitted by the authors in response are acceptable.

Minor:

Legend S5:

"At night" would be more accurate to say "In the dark".

Response: Agreed and changed in revised legend for figure 4 (formerly S5).

REVIEWERS' COMMENTS:

Reviewer #3 (Remarks to the Author):

The authors have tried to address this reviewer's concern about whether UBP12C208S 180 -YFP is as stable as UBP12-YFP. Apparently transgene silencing has prevented comparative immunoblot determination of the steady state accumulation of the two forms of the tagged protein. As an alternative they transiently overexpress (35S promoter) GFP-tagged versions of both proteins in *N. benthamiana* and show by quantitative imaging of the nuclei that both forms accumulate to a similar level.

While this is less than ideal, due to the possibility that strong overexpression overwhelms the natural degradation process of the UBP12 protein, this appears to be among the best that can be accomplished at this point. An alternative would be *in vivo* degradation assays in these transient transformants, with CHX applied to inhibit translation, and then observe the time course of loss of the two protein species. This type of experiment has been reported in plant research many times, though usually testing in planta under native promoter expression. This would assume that the degradation mechanism in *N. benthamiana* is similar to that in *Arabidopsis* and still would have the complication of strong overexpression.

Under the circumstances and in the context of the greater message that is supported by other data, this reviewer accepts the alternative approach they have chosen and reported.